**communications** engineering

# Additive manufacturing of a 3D-segmented plastic scintillator detector for tracking and calorimetry of elementary particles
Tim Weber[1], Andrey Boyarintsev[2], Umut Kose[1], Botao Li[1], Davide Sgalaberna [1] ✉, Tetiana Sibilieva [2], Johannes Wüthrich[1], Siddartha Berns[3,4,5], Eric Boillat[3,4,5], Albert De Roeck [6], Till Dieminger[1], Matthew Franks[1], Boris Grynyov[2], Sylvain Hugon[3,4,5], Carsten Jaeschke[1] & André Rubbia[1]

Plastic scintillators, segmented into small, optically isolated voxels, are used for detecting elementary particles and provide reliable particle identification with nanosecond time resolution. Building large detectors requires the production and precise alignment of millions of individual units, a process that is time-consuming, cost-intensive, and difficult to scale. Here, we introduce an additive manufacturing process chain capable of producing plastic-based scintillator detectors as a single, monolithic structure. Unlike previous manufacturing methods, this approach consolidates all production steps within one machine, creating a detector that integrates and precisely aligns its voxels into a unified structure. By combining fused deposition modeling with an injection process optimized for fabricating scintillation geometries, we produced an additively manufactured fine-granularity plastic scintillator detector with performance comparable to the state of the art, and demonstrated its capabilities for 3D tracking of elementary particles and energy-loss measurement. This work presents an efficient and economical production process for manufacturing plastic-based scintillator detectors, adaptable to various sizes and geometries.

Plastic scintillator (PS) detectors, invented in the early 1950s[1], are widely used in the detection of elementary particles in high-energy physics[2–6], nuclear physics[7], astroparticle physics[8,9], as well as in many applications like muon tomography[10], proton computed tomography for hadron therapy[11], fast-neutron detection[12,13] and non-destructive imaging[14]. By measuring the energy loss of a particle and tracking its path in the detector, it is possible to identify the type of interacting particle, reconstruct its momentum based on range, measure its original energy using calorimetry, and determine its electric charge if the setup is immersed in a magnetic field. Another important feature of PS detectors is their unique ability to provide a fast response, with time resolution in the sub-nanosecond range.

In the context of high-energy physics, PS detectors are typically made of long scintillating bars with O(cm) granularity for time of flight detectors[15,16], neutrino active targets with tonne-to-kilotonne scale mass[2–4],

sampling calorimeters made by layers of segmented PS alternated with heavier materials like iron and lead[17], or scintillating optical fibers[5] with a diameter down to 250 μm. A PS is an organic material composed of a mixture of carbon and hydrogen-based molecules, usually polystyrene (PST) or polyvinyltoluene (PVT). Molecules of an activator like p-terphenyl (pTP), 2,2-p-phenylene-bis(5-phenyloxazole) (POPOP), 2,5-diphenyloxazole (PPO) are introduced into the polymer matrix at a level between a few per mille and percent by weight.

The scintillation mechanism consists of a few steps, as illustrated in Fig. 1a. When a charged particle propagates through a PS, the molecules of the polymer matrix get excited. A fast, short-range resonant non-radiative dipole-dipole interaction, called Förster mechanism[18], efficiently transfers the excitation energy to the activator, which de-excites and emits near-ultraviolet (UV) photons with minimal emission delay. A second dopant,

[1]Institute for Particle Physics and Astrophysics, Federal Institute of Technology Zurich (ETH), Zurich, Switzerland. [2]Institute for Scintillation Materials (ISMA), National Academy of Sciences of Ukraine, Kharkiv, Ukraine. [3]Haute Ecole Spécialisée de Suisse Occidentale (HES-SO), Delémont, Switzerland. [4]Haute Ecole d'Ingénierie du canton de Vaud (HEIG-VD), Yverdon-les-Bains, Switzerland. [5]Technopole de Sainte-Croix, COMATEC-AddiPole, Sainte-Croix, Switzerland. [6]Experimental Physics Department, European Organization for Nuclear Research (CERN), Geneva 23, Switzerland. ✉e-mail: davide.sgalaberna@cern.ch

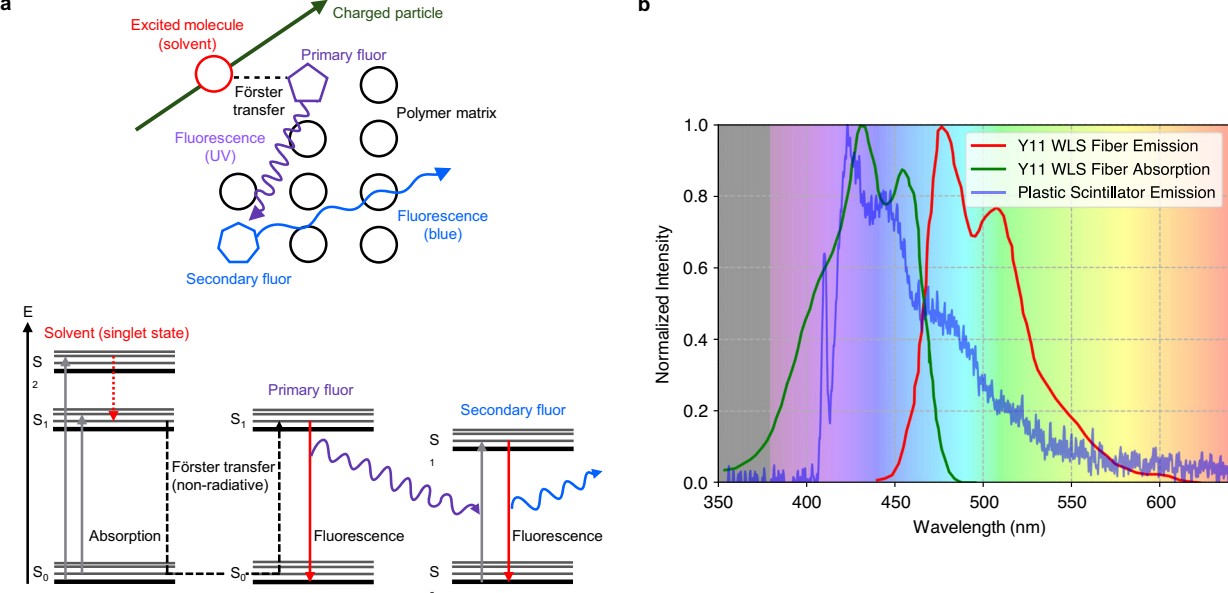

**Fig. 1 | Scintillation process** Absorption and emission spectra of detector components (**a**) in a plastic scintillator matrix, doped with primary and secondary fluors, the scintillation process is initiated when a charged particle passes through the material. **b** Emission spectrum in blue of a 3D-printed plastic scintillator sample recorded with a spectrophotometer, absorption spectrum in green, and emission spectrum in red of a Kuraray Y11 wavelength-shifting fiber. The background color represents the light's color as a function of wavelength.

called a shifter, is usually added to change the wavelength of the light to avoid absorption in the material[19,20]. In high-energy physics experiments, the light produced in PS is frequently collected using wavelength-shifting (WLS) fibers. These fibers shift the light from the violet/blue spectrum, which is the typical emission range of PS, to green, where the attenuation length is longer. Then, they guide it toward photodetectors, taking advantage of the fiber's long attenuation length, which spans over several meters[2].

PSs for high-energy physics are traditionally manufactured using: (1) cast polymerization[21], where a heated liquid monomer with dissolved dopants is poured into a mold, that results in a solid plastic structure after cooling; (2) injection molding[22,23], where polymer granules compounded with dopants are melted and mechanically forced into a mold to solidify; (3) extrusion[24,25], which pushes melted plastic through a die of the desired cross-section. While cast polymerization provides the best optical properties, the more commonly used injection molding and extrusion offer a simpler, faster, and cheaper production, hence optimal for large-volume detectors[25]. PS detectors composed of scintillating voxels, optically isolated from each other, are optimal for particle tracking. The typical light reflector is composed of a white diffuser, such as paint containing titanium dioxide, and white polytetrafluoroethylene (PTFE), or is created using a process of chemical etching[26]. Co-extrusion of PS and optical reflector has also been conducted[2,4]. Another option with good performance consists of wrapping each scintillating element with Tyvek[27]. However, such a solution is unfeasible for detectors with a large number of PS elements[28–30].

In recent years, there have been advancements in the development of novel three-dimensional (3D) granular scintillating detectors for imaging electromagnetic and hadronic showers[31], as well as neutrino interactions[28–30,32]. Combining these geometries with the PS sub-ns response is ideal for efficient neutron detection with kinetic energy reconstruction by time of flight[33–36]. For instance, a neutrino detector made of two million PS cubes, each of size 1 cm³, with a total active mass of two tonnes, has been proposed[28], prototyped[29,30], built, and started collecting data as of 2023 at the T2K neutrino experiment in Japan[37,38]. Simulation studies conducted independently on a comparable 3D granularity plastic scintillator detector demonstrated that a tracking resolution ranging between 2 mm and 4 mm,

depending on the utilized reconstruction algorithm, can be attained for minimum ionizing particles such as muons[39]. The complexity in the geometry of such a detector requires multiple vastly different manufacturing steps including the fabrication of every single PS cube, optical isolation via chemical etching, and drilling of the holes for WLS fiber placement. Additionally, the assembly of two million cubes demands a considerable effort[40] and needs to be combined with a robust box that mechanically maintains the structural integrity of the entire construction[37]. An attempt to simplify detector fabrication included the development of a prototype of PS cubes glued together, achieving a tolerance of approximately 200 μm[41]. However, such a method is not feasible for the production of a single 3D volume of PS cubes, but only of 2D layers, whose production would also be time-consuming for large-scale applications.

The above considerations call for the development of a novel manufacturing process that allows for the easy production of several thousand optically isolated PS cubes in a single block of plastic. Our solution takes advantage of additive manufacturing (AM), which opens the door to automated processes that could simplify the construction of future particle detectors. More commonly known as rapid prototyping or 3D printing, AM processes build the designed parts through a layer-by-layer addition of new material. AM technologies are capable of fabricating customizable, mono-lithic parts with multiple materials and complex internal geometries, and can reduce production time and cost. Two common AM technologies for polymers are: stereolithography[42], where liquid resin is solidified after curing with, for example, UV light; and Fused Deposition Modeling (FDM)[43], where, analogous to extrusion, a thermoplastic material in the form of a thin wire is passed through a feeding into a melting system and deposited on a print bed line-by-line with the help of a nozzle tip[44,45]. 3D printing of PS using stereolithography has been reported in the literature. However, studies have shown the necessity of either developing a new chemical composition or binding PS granules into a polymer matrix. This poses a challenge in achieving competitive performance levels in terms of acceptable light yield and attenuation length compared to standard PS[46]. In recent years, progress has been made in the development of curable resins, primarily targeting applications beyond high-energy physics. These advancements have demonstrated good performance in terms of light output, pulse shape

discrimination, thermal neutron sensitivity, and various other properties when compared to commercially available alternatives[47–52].

Anyhow, it is difficult to achieve 3D printing of multiple materials, along with the production of hollow objects that possess smooth and consistent inner surfaces, which is needed for the insertion of WLS fibers. A promising additive manufacturing option is FDM, which allows the use of the same chemical composition as standard scintillators in the form of a filament, ensures high transparency and multi-material printing, and has been successfully used to 3D print PST-based scintillators[53]. Another independent work reached a similar conclusion[54]. A technical attenuation length of about 19 cm, measured in a 5 cm long bar, was later demonstrated, which is sufficient for fine-granularity scintillator detectors[55]. For such a scintillator composition, the emission spectrum of the 3D-printed PS peaks around 420 nm upon the excitation with 386 nm light (Fig. 1b). This result is consistent with the UPS-923A standard scintillator[56,57]. It also shows that the measured scintillation spectrum closely matches the absorption spectrum of green WLS fibers such as the Kuraray Y11[58], allowing for an efficient readout of the scintillation light. Thereafter, the first 3D-printed matrix of PS cubes optically isolated with a custom-fabricated white reflective filament (PST, or polymethyl methacrylate (PMMA)) mixed with titanium dioxide was produced and showed a low cube-to-cube light crosstalk, less than 2%[55]. However, after 3D printing, no work reported in literature could avoid the use of subtractive processes, like polishing the outer surface to achieve a good geometrical tolerance. Moreover, none of the studies ever attempted to 3D print hollow PS objects to host WLS fibers.

In this work, we demonstrate the additive manufacturing of a 3D-segmented, fine-granularity PS detector without the need for any post-processing. A SuperCube, consisting of a $5 \times 5 \times 5$ matrix of 1 cm$^3$ optically isolated cubes, was additively manufactured and tested. Each cube included cylindrical holes to house WLS fibers throughout the entire detector.

## Results

Previous R&D on this topic did not yield a particle detector of sufficient quality[53,55], as described in the Introduction. However, it was useful in defining the extrusion temperature of the PS, at which there is no loss of the original scintillation light yield, and it showed the producibility of a well-performing optically reflective filament.

In this work, a manufacturing method named Fused Injection Modeling (FIM) was developed to obtain good geometrical tolerances, high transparency PS volumes, as well as precise hole fabrication for the placement of WLS fibers at a rapid production speed, thereby overcoming the aforementioned shortcomings of the AM fabricated PS particle detector. FIM merged the geometrical freedom of manufacturing FDM with the production speed and high part density of injection molding by 3D printing an optically reflective frame containing the desired voxel shape and quantity, which is filled by rapidly pouring melted PS into the empty cavities to accurately shape the geometry of the PS (detailed in the "Particle detector fabrication" section).

To achieve a fast, high-quality, consistent forming of the PS structure, a customized liquefaction system integrated into an FDM machine was developed. Standard FDM melting components were analyzed and modified to cope with the increased thermal demands of a rapid heat transfer toward the low thermally conductive PS while guaranteeing a working temperature in the filament-feeding parts of the extrusion system. To distribute scintillation material evenly throughout the entire volume, an elongated nozzle was manufactured that provided freedom of movement within the already fabricated cavity. It was coupled with a spring-pressurized plate that constrained the melt pool within its mold while allowing air to escape the volume during the forming process. A more in-depth explanation of the custom-made extrusion system can be found in the "Plastic scintillator forming" section. A Computational Fluid Dynamics (CFD) analysis was performed to determine the material requirements of the melting components, heat block, and nozzle; the heat shielding parts, feeding tube, and heat break; the process parameters, heat block temperature, extrusion speed, and polymer temperature at the nozzle orifice. A detailed explanation

of the nomenclature of the machine components, design, and mode of operation of the manufacturing process is found in the "Particle detector fabrication" section.

The results of the CFD analysis concluded that both the heat block and the nozzle needed to be manufactured from copper. Its high thermal conductive property enables a rapid heat transfer to the low thermally conductive PST-based PS filament, such that the desired melt temperature can be reached at high extrusion speeds. The feeding tube had to be fabricated with polyether ether ketone (PEEK), the heat break from stainless steel with a wall thickness of 250 μm. The low thermal conductivity of these materials combined with the small cross-section of the heat break resulted in a highly thermal resistant structure that prohibits heat flow toward the temperature-sensitive parts and maintains a working temperature of the whole extrusion system. This composition of components leads to optimal process parameters: a maximum throughput speed of 15 mm s$^{-1}$, to generate a high mass flow that quickly spreads within the cavity before solidifying; at a peak heat block temperature of 300 °C, with which is premature melting followed by system clogging can be avoided. This combination resulted in a temperature of around 230 °C throughout the entire cross-section of the PS at the nozzle orifice, which preserves the scintillation properties established in previous studies[53,55]. Figure 2a illustrates the components of the CFD analysis, showing a steep temperature gradient generated by the heat break, which reduces heat transfer upward, maintaining a safe working temperature. It also shows the transition of the plastic scintillator as it flows downward, gradually melting inward and reaching an exit temperature of approximately 230 °C.

The performance of the custom heat break was tested by measuring its temperature with a thermocouple at a heat block temperature of 300 °C and no filament extrusion. The results showed a 43.2% temperature reduction from 155 °C to 88 °C compared to a standard model heat break. This value lies below the glass transition temperature of the PS (100 °C), thus ensuring a failure-free extrusion process.

The fabrication procedure used the custom-manufactured modified extrusion system combined with a spring-loaded plate that enabled the elongated nozzle to move within the cavity and constrained the melt pool at the end of the filling process (Fig. 2b). The procedure involved three main steps: the fabrication of one matrix layer of the reflective frame via FDM (Fig. 2c); the insertion of metal rods into the voxel cavity through already 3D-printed holes in the frame; followed by the forming of the cube-shaped volume with melted scintillation material (Fig. 2d). The rods were essential, as WLS fibers could not be positioned during the PS forming process without risking thermal damage. These processes can be repeated until a SuperCube of the desired size has been obtained. Finally, the WLS fibers were placed through the PS voxel via the cylindrical holes produced by the removed metal rods (Fig. 2e).

The optically reflective frame constituted the mold, with which the PS was formed and thus needed to withstand the pressure and heat of the extruded melt pool. As reported in our previous work[55], in order to manufacture a 3D-printed layer of optically isolated cubes, a custom optical reflector filament was fabricated with either PMMA or PST. Although a transmittance of less than 10% for a 1 mm thickness and an optical cube-to-cube light crosstalk of less than 2% were achieved, the material could not maintain its structural integrity during the injection of the PS. This was due to the similar heat resistance of the optical reflector compared to the PS, with the consequence of swelling and wall bending during filling. Hence, for the FIM method, a white, more heat-resistant polycarbonate-PTFE filament was used, which combines good optical properties of PTFE and a heat resistance higher than the previously used materials. In FDM, vertical walls are created by stacking lines on top of each other, while horizontal walls are formed by placing adjacent lines on the same print plane. This results in different fill factors, thus different light transmissive properties. The corresponding transmittance for a wavelength of 420 nm, measured with a monochromatic light source in air, resulted in 13% for horizontally and 18% for vertically built walls (Fig. 3). Consequently, to obtain a uniform cube-to-cube light crosstalk, horizontal walls were fabricated with a thickness of

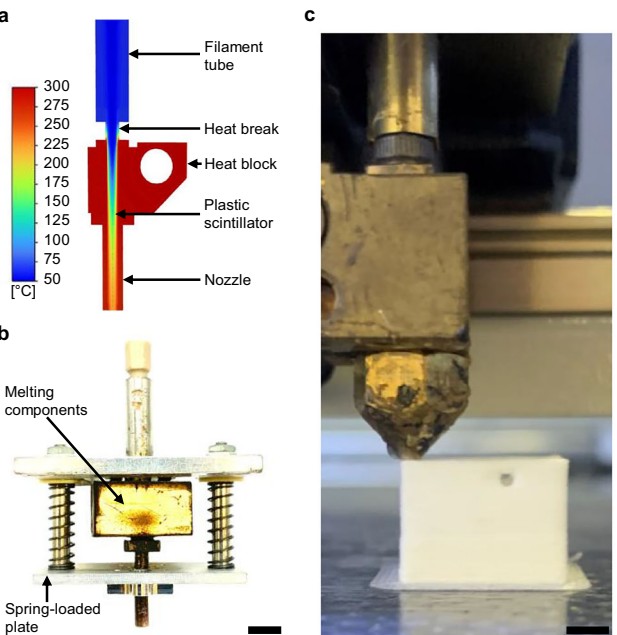

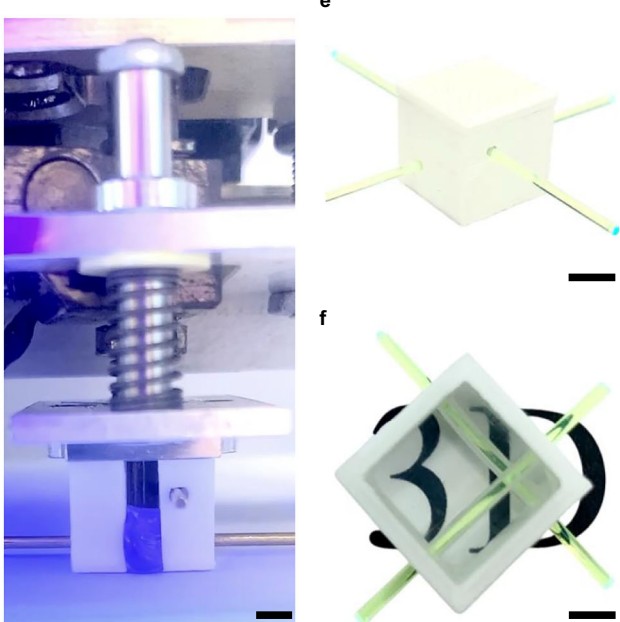

**Fig. 2 | Manufacturing process of an optically isolated plastic scintillator voxel.** **a** CFD result of the injection system at an extrusion speed of 15 mm s⁻¹. Flow direction from top to bottom. The color scheme represents the temperature in °C at specific locations in the extrusion system. **b** Injection system used to form PS within an FDM-fabricated reflective frame. Scale bar, 6.5 mm. **c** FDM fabrication of a reflective frame with holes for the positioning and insertion of WLS fibers. Scale bar, 3 mm. **d** Forming of PS volume. To better visualize the flow of melted PS, the sample was illuminated with UV light, and the cavity was opened to provide a clear view. Scale bar, 3 mm. **e** Completed optically isolated PS voxel equipped with WLS fibers. Scale bar, 6 mm. **f** A voxel without top and bottom layer. The bottom face of the PS was polished to remove the surface roughness left by the FDM-fabricated mold during the injection process, allowing the transparency of the scintillator to be showcased. Scale bar, 3 mm.

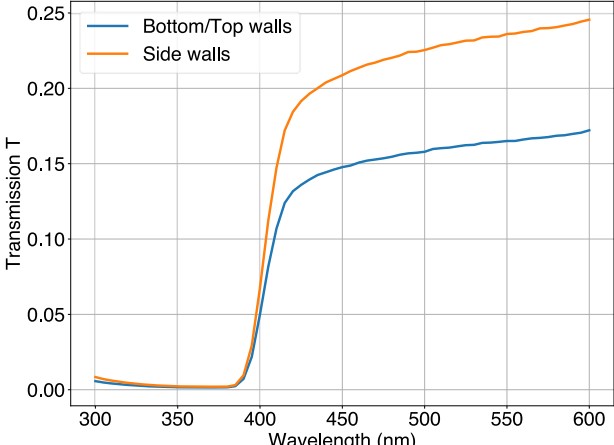

**Fig. 3 | Light transmittance of the reflector material.** Light transmittance measurements of white reflector sheets with a thickness of 1 mm. Bottom and top walls in the SuperCube were built horizontally, side walls were built vertically.

1.2 mm and the more transmissive vertical walls with 1.5 mm. The more heat-resistant reflective material retained its as-built cube shape throughout the filling fabrication of the PS, as can be seen in (Fig. 2f). Caliper measurements of vertical walls showed an average thickness of $t_{mean} = 1.51$ mm, deviating by only 0.01 mm from the nominal $t_{designed} = 1.5$ mm with a standard deviation of $t_{std} = 0.01$ mm. Both the retention of the reflective frame geometry after filling and the accurate fabrication of the wall thickness ensured a consistent active volume of scintillation material in every voxel. More details about the fabrication of the white reflector walls can be found in Methods 'Reflective Frame Fabrication'.

The established process parameters combined with the custom-manufactured elongated nozzle and melt-pool-containing pressurized plate

resulted in a transparent PS volume with a high fill factor around the metal rods, which left precisely positioned and close-fitting holes for the insertion of WLS fibers. In FDM-manufactured cubes, a technical attenuation length of about 19 cm was measured and found to be sufficient for highly segmented detectors[53]. The FIM-fabricated PS showed an even improved transparency and no air bubbles compared to the samples produced with FDM (Fig. 2f).

The manufacturing time per voxel, as depicted in Fig. 2e, was approximately 6 min. This involved the fabrication of the reflective frame and the forming of the PS. Fixed time cost, including the not-yet fully automated change between the FDM and injection setup, was around 20 min, plus an additional 15 min for an initial warm-up of the machine.

The $5 \times 5 \times 5$ voxel SuperCube, obtained using the FIM method without the need of post-processing, is shown in Fig. 4a, b. It was instrumented with WLS fibers (along the X and Y axes) that capture and guide the scintillation light produced by charged particles in single cubes toward coupled silicon photomultipliers (SiPM). These count the number of photoelectrons (PE), which are the primary electrons generated via the photoelectric effect by the visible photons impinging on the SiPM's active area. The instrumented detector is shown in Fig. 5a. More details about the detector setup can be found in the "Detector setup and analysis method for cosmic data" section.

The response of the SuperCube was characterized by cosmic particles in terms of single-cube scintillation light yield and cube-to-cube scintillation light crosstalk, which are the main parameters that determine the detector performance in terms of particle tracking, identification, and calorimetry. Particles producing a vertical track, whose range in the detector can be more precisely estimated, were selected and used for analysis. Being mostly minimum ionizing particles, their typical energy deposited in PS is approximately 1.8 MeV cm⁻¹, which allows us to precisely estimate the number of photons detected per-unit energy loss. Moreover, with an energy spectrum that spans from a few hundred MeV to several GeV, cosmic

**a**

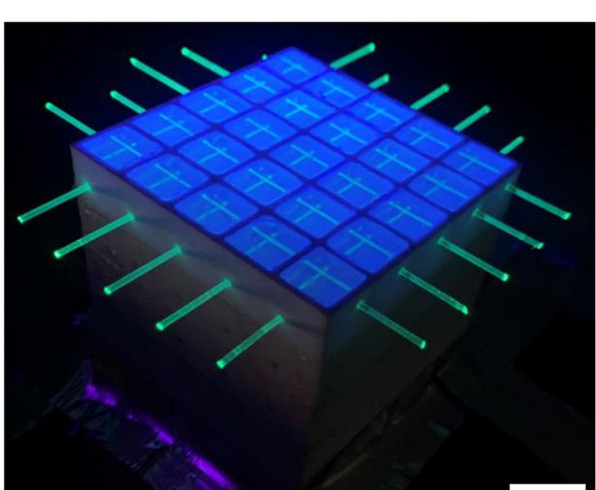

**b**

**Fig. 4 | FIM-fabricated SuperCube. a** The fifth layer of the SuperCube, unsealed and illuminated with UV light, showcasing the active PS volume within each voxel and the WLS fibers extending through the entire detector. Scale bar, 14 mm. **b** Completed FIM-manufactured 5 × 5 × 5 voxel SuperCube. Scale bar, 15 mm.

**a**

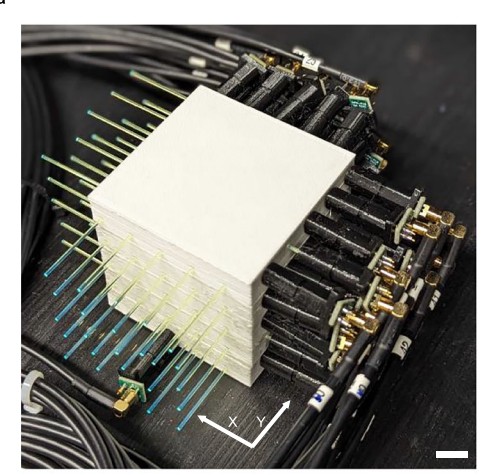

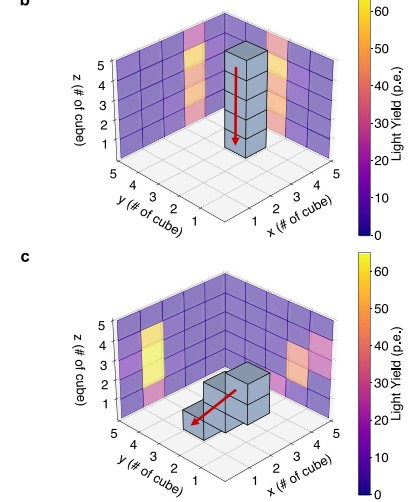

**Fig. 5 | Cosmic particle tracking. a** FIM-manufactured SuperCube instrumented with WLS fibers and silicon photomultipliers in the X and Y directions. Scale bar 10 mm. **b** Path reconstruction of a cosmic particle traversing the SuperCube along a vertical trajectory. **c** Path reconstruction of a cosmic particle traversing the SuperCube along a diagonal trajectory. In both (**b**) and (**c**), the grid in the X, Y, and Z directions represents the voxels within the SuperCube, while the color scheme in the 2D projections represents the measured light yield in units of photoelectrons detected in each readout channel.

particles typically produce through-going tracks in the detector with a very uniform energy deposition. By measuring the number of PE across all the readout channels, and geometrically matching the two detector projections of the event, the 3D track of the particle can be reconstructed and the number of scintillation photons produced in each cube can be accurately measured. Figure 5b, c shows the events of cosmic particles crossing the SuperCube from top to bottom. One particle follows a vertical track (Fig. 5b), while the other follows a diagonal path (Fig. 5c). The 2D projections display the number of PE detected in each readout channel (X and Y), and the 3D voxels illustrate the reconstructed particle tracks, each highlighted by a red arrow.

Figure 6a shows the light yield distribution across all voxels in the SuperCube and Fig. 6b the cube-to-cube crosstalk distribution throughout the entire SuperCube. The scintillation light yield of the 3D-printed prototype was found to be comparable to that obtained with the one produced by cast polymerization (using the same scintillator composition)[41]. A most probable value of about 29 PE was measured for the 3D-printed SuperCube,

close to the most probable value of the prototype produced with cast polymerization (about 28 PE).

In summary, the newly developed FIM method allowed for the production of a monolithic 3D optically segmented 5 × 5 × 5 matrix of scintillating cubes each measuring 1 cm³. These cubes were fabricated with accurately placed holes for the insertion of WLS fibers in both the X and Y directions. No post-processing was necessary and the detector could be instrumented with the read-out electronics right after the fabrication process was completed. Hence, a PS detector capable of tracking elementary particles and measuring their energy loss has been successfully additively manufactured. Experimental tests showed a detection performance comparable to the state-of-the-art detectors produced with traditional manufacturing techniques.

## Discussion
The FIM fabrication process has made it possible to capture a 3D image of particle interactions using a particle detector entirely produced through

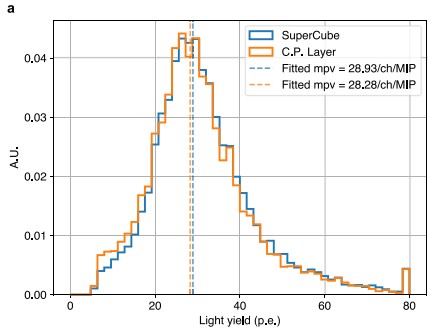
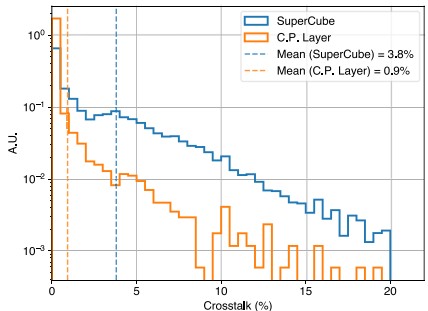

**Fig. 6 | SuperCube performance. a** Scintillation light yield distribution across all voxels in the SuperCube in blue, and the cast polymerization layer (C.P. layer) in orange. **b** Scintillation light cube-to-cube crosstalk distribution throughout the entire SuperCube in blue and the cast polymerization layer in orange. Both, in (**a**) and (**b**), the values were normalized to the number of events recorded during the measurement period.

additive manufacturing. The development of this method was important for achieving the improved results described above, as it brought several advantages over FDM and other more standard methods like injection molding or cast polymerization.

The FDM-manufactured PS showed sufficient transparency for centimeter-sized voxels. However, the FIM-fabricated samples visibly improved the quality by minimizing imperfections created by the merger of different deposited lines during the FDM process and by reducing air bubbles within the PS volume. The improved transparency results from the straight down-to-up filling pattern, which created a single melt pool that expanded and formed the PS structure within the optical reflective frame. The nozzle's orifice remained submerged in the melt pool until the PS forming process was complete. This technique prevented air from being trapped between different layers of melted polymer. Additionally, the forming of a cubic-centimeter-sized PS cube with FIM needed around 30 s, whereas the same structure took around 5 min using FDM, about ten times longer. The faster per-unit manufacturing time is crucial when it comes to the production of large-volume, highly segmented particle detectors that can include millions of isolated voxels.

The optical isolation of PS voxels is included in the FIM process via the FDM-manufactured reflective frame. This is superior to traditional methods, which require additional vastly dissimilar fabrication steps, as described in the Introduction. The sequence of different fabrication processes increases the total time and cost of production due to the involvement of multiple manufacturing parties, transportation, and dead time between steps. The geometrical tolerance of the optically isolating frame produced by FIM is improved compared to the particle detector manufactured solely by FDM. This is a result of the introduction of a heat-resistant material that was able to preserve its designed shape during the filling of melted PS, thus creating a distinct boundary between the individual PS voxels. In FDM, both the scintillation and the reflective filament are made of plastics with similar thermal characteristics, which can result in a mixing at the boundary layers between these two materials and also a warping and distortion of the reflective geometry. Although the tolerance cannot be considered as good as the one achievable with injection molding or cast polymerization (better than 50 μm for plastics), it is worth noting that the quality is affected by the type of 3D-printed material, the printing strategy and the precision of the machine, which will be improved in future studies.

The primary advantage of the FIM method compared to FDM or more traditional methods is the capability of creating precise, close-fitting holes through the entire particle detector to host WLS fibers. FDM is notorious for producing poor-quality small holes in terms of size consistency, shape, and surface roughness, which results in a low WLS fiber to PS contact area, especially perpendicularly to the print plane. Injection molding has the capability to create hollow components within the manufactured structure. However, incorporating a white reflector without damaging the pre-existing holes poses a considerable challenge. In the next prototype, a third vertical hole will be added. An alternative option would be to utilize thin pipes capable of withstanding temperatures much higher than 300 °C, in place of metal rods. This modification would facilitate the easy insertion of WLS fibers, leading to the creation of the SuperCube. A single-layer prototype was successfully produced using borosilicate glass pipes with outer and inner diameters of 1.5 mm and 1 mm, respectively. While this approach eliminated the need to extract rods after the injection and solidification of the plastic scintillator, the addition of non-plastic material does slightly increase the inactive volume. Consequently, we opted to prioritize the baseline option involving metal rods.

In terms of the detection performance, the quantity of scintillation light, collected from the FIM-manufactured PS cubes crossed by cosmic particles, is similar to that observed from PS cubes produced with traditional methods like cast polymerization. The cube-to-cube crosstalk at a few percent level is comparable to that of typical PS detectors[30] and is low enough to provide unambiguous 3D particle tracking. The light crosstalk is about four times higher compared with the cast polymerization prototype, due to the higher wall transmittance. Hence, one would expect an approximately 20% lower total light yield. However, this was not observed, as shown in Fig. 6a. One may conclude that the additive manufactured PS has an intrinsic light output higher than the one produced by cast polymerization, but this was not the result of our past work[53,55]. It is well known that the light yield of a PS detector that uses WLS fibers can be enhanced by improving the light trapping efficiency of the WLS fibers. This can be achieved by reducing air gaps, thereby increasing the contact area between the PS and the WLS fibers, whose similar refractive indices allow a high transmittance from the scintillation material to the fiber. This can be obtained by using optical grease[59–61], which is difficult to homogeneously introduce in long, small-diameter holes. Using the FIM method, it was possible to create holes, that span the length of the SuperCube, only 100 μm larger than the WLS fibers, resulting in a large contact surface between the PS cubes and the fibers. This feature is expected to increase the total light yield and compensate for the light loss from the observed crosstalk.

A preliminary time resolution measurement was conducted using a Hamamatsu H6410 photomultiplier tube coupled to a sample of a 3D-printed plastic scintillator exposed to a $^{60}$Co radioactive source. The data was read out with a LeCroy Wave runner 104MXi-A oscilloscope, revealing a decay time consistent with that of UPS-923A, indicating no degradation of the timing properties resulting from the scintillator additive manufacturing process. However, a more comprehensive quantitative characterization is deferred to future work and publication.

Aging presents another concern, potentially resulting in a few percent reduction in light yield annually for extruded polystyrene-based scintillators[56,62–64]. Given the similarity of the FIM process with extrusion or injection molding in the treatment of the plastic scintillator material, a similar degradation is expected. Despite various measurements conducted over approximately four months, no visible changes in light yield and crosstalk were observed within the precision of the experimental setup.

Nevertheless, more precise quantitative results require dedicated aging measurements, which are planned for future investigation.

Overall, the FIM method resulted in a faster particle detector fabrication by including the shaping of the PS in optically isolated voxels into an already assembled final detector using one single machine. From here, we see no obstacles that would impede the viability of this method for large-scale particle detector production. Each manufacturing step has consistently yielded satisfactory results.

In conclusion, we have successfully achieved the additive manufacturing of a 3D optically segmented plastic scintillator detector, capable of both particle tracking and energy-loss measurement. FIM demonstrates a tool that enables new concepts for particle detectors, characterized by a very fine 3D granularity in very large active volumes, above the tonne-scale. Although not tested in this work, millimeter-sized granularity and bigger units, e.g. $30 \times 30 \times 30$ cm$^3$ or above, are expected to be achieved. We believe that this technology opens new possibilities for the future of particle physics experiments. In particular, long-baseline accelerator neutrino oscillations[65–68] or those searching for new neutrino sterile states via short-baseline oscillations[69–74] can profit from the developed additive manufacturing process by building very large but finely segmented scintillator detectors to obtain a large sample of very detailed images of neutrino interactions. Moreover, this production method can make it easier to build highly performing fine-granularity electromagnetic or hadronic sampling calorimeters, enabling high-resolution particle flow analysis and fulfilling the requirements of future collider experiments[31,75]. FIM can be performed with various types of filaments, where the white reflector could be covered by an additional layer made of heavier nuclei or doped material. Finally, large-volume 3D detector geometries, made possible by the FIM process, can enable a high-efficiency detection of fast neutrons with a precise measurement of their time of flight, thanks to the large content of hydrogen in PS and its short decay time[28,35]. This is of fundamental importance for next-generation neutrino physics experiments, providing constraints on leading systematic uncertainties[33,34,76,77].

## Methods
### Particle detector geometry
The overlaying structure of the particle detector was a cuboid that consisted of smaller, cube-shaped detection units called voxels. One voxel consisted of three elements: a $10 \times 10 \times 10$ mm$^3$ PS volume for visible light production; a reflective frame with a wall thickness of 1.5 mm in the and 1.2 mm in the horizontal plane direction, enclosing the PS to entrap the light in the unit; and two perpendicularly arranged WLS fibers in the horizontal plane with the coordinates (Y = 2.5 mm, Z = 7 mm) in X-direction and (X = 2.5 mm, Z = 3 mm) in Y-direction that absorbed the light from the PS, shifted its wavelength such that the event could be read out by an external system connected to the fibers (Fig. 7a). The SuperCube prototype detector consisted of 125 detection units arranged in a $5 \times 5 \times 5$ configuration, resulting in total dimensions of 59 mm in width and 57.2 mm in height, as illustrated in Fig. 7b.

### Particle detector fabrication
The SuperCube was produced in two separate processes; the fabrication of a $5 \times 5$ matrix layer of an optically reflective frame and the forming of the PS, each using a different mechanical setup. The two production steps were alternated to build a layer-by-layer three-dimensional $5 \times 5 \times 5$ SuperCube. The reflective frame with holes for the WLS fibers was manufactured via FDM without the use of a support structure (Fig. 8a). The PS was formed in three sub-processes. First, metal rods were placed through the holes to create circular voids for the WLS fibers to be inserted, then the square-shaped volume was rapidly filled in a bottom-to-top motion with a customized extrusion setup combined with a pressurized plate to keep the melt constrained in the cavity (Fig. 8b) and thirdly, a heated punch was pressed on top of the PS to plane its surface such that the next matrix layer could be fabricated on top of it (Fig. 8c). Finally, the

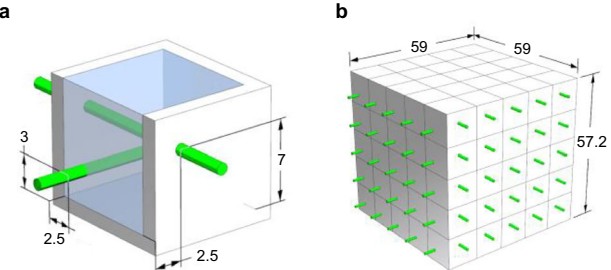

**Fig. 7 | SuperCube geometry. a** Components of a voxel with sections of the reflective frame cut away on the top and side. The reflective shell in white encloses the cube-shaped PS in blue, which is traversed by two wavelength-shifting fibers in green positioned at their respective coordinates in millimeters. **b** Depiction of a $5 \times 5 \times 5$ voxel SuperCube with its outer dimensions in millimeters.

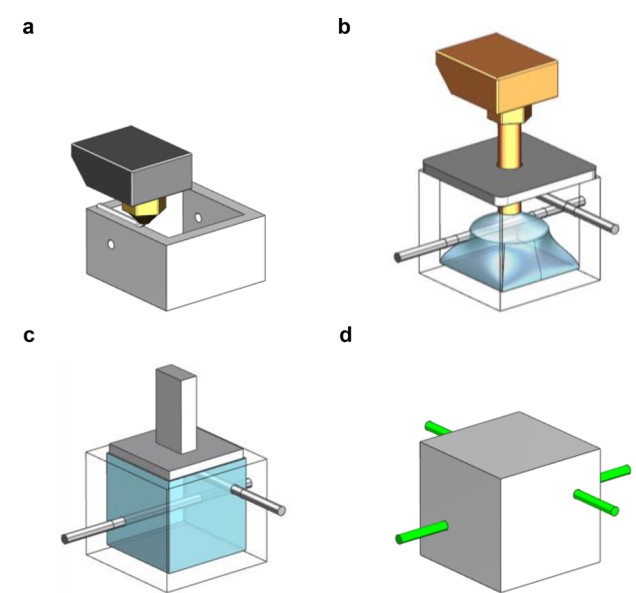

**Fig. 8 | Depiction of the FIM fabrication process. a** Reflective frame fabrication via fused deposition modeling. **b** Plastic scintillator forming using a customized extrusion setup. Metal rods in black create circular voids for wavelength-shifting fiber insertion. The pressurized bracket on top of the voxel constrain the melt pool within the cavity. **c** Planing of the top surface using a heated punch. **d** A completed voxel with wavelength-shifting fibers in green inserted through the whole structure.

top of the voxel was closed by a top layer and the WLS fibers were inserted through the already manufactured holes (Fig. 8d).

**Reflective Frame Fabrication**. The reflective shell was manufactured via FDM using a Creatbot F430[78]. In FDM, a polymer string is processed through an extrusion system, which is divided into a cold and hot zone, enabling the distribution of the material with a predetermined temperature and shape. The role of the cold zone is to push the filament with the appropriate speed through feeding tubes into the hot zone, using a motor-controlled gear and roller structure. In this part of the extrusion procedure, the machine temperature must be below the glass transition temperature of the polymer-in-fabrication, otherwise, premature softening of the string leads to an increase in friction with the feeding tubes, followed by clogging and system failure. The purpose of the hot zone is to rapidly melt the low thermally conductive material through a metal heat block kept at a polymer-specific temperature, such that it can be distributed via an extrusion nozzle into the desired shape. The link between both zones is the heat break, a thin circular tube, which acts as a thermal

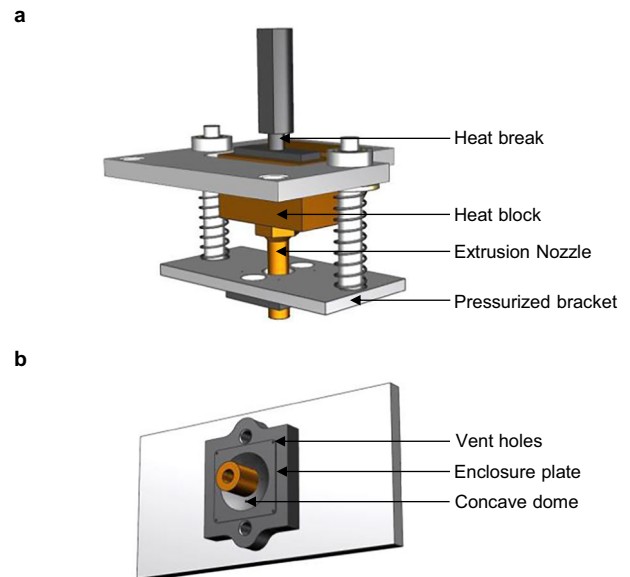

**Fig. 9 | FIM injection components. a** Depiction of the extrusion system components modified for the filling of scintillation material into the reflective cavity. **b** Geometry of the pressurized plate that is pressed onto the cavity to keep the melt pool restrained in fill volume.

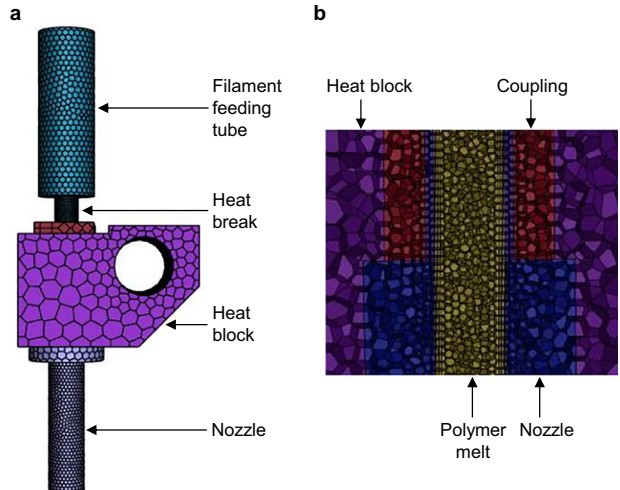

**Fig. 10 | CFD analysis setup. a** Extrusion components of the CFD model. Each component is assigned a unique color, while the black mesh represents the discretization points of the structure. **b** Cross-sectional view of the interior of the model, depicting the polyhedral volume mesh.

resistor. It prohibits the heat from creeping from the hot into the cold zone, thus establishing a working temperature in both areas of the extrusion system.

A 1.75 mm polycarbonate-PTFE-blend filament from Rosa3D filaments[79] was used for the frame fabrication. This material provided the required combination of good optical reflectivity to contribute to the performance of the detector and high thermal resistance to ensure the dimensional integrity of the geometry during the forming of the PS.

The structure was printed with a nozzle temperature of 280 °C, a bed temperature of 115 °C for the first matrix layer, and a chamber temperature of 55 °C to reduce the risk of warping. After the first filling cycle, the bed temperature was set to 75 °C at all times to avoid permanently turning the scintillation material opaque, as was observed in previous studies[55]. The reflective structure was fabricated using a 0.4 mm diameter nozzle with a fill factor of 100% for both horizontal and vertical walls. The layer height was set to 0.2 mm, the print speed to 2000 mm min$^{-1}$, and cooling was activated only during the layers incorporating circular holes to ensure accurate round geometries without the need for support structures. In FDM, the thickness of horizontally printed walls (parallel to the print bed) is very close to the designed value and was not further investigated. However, 20 thickness measurements of vertically printed walls (perpendicular to the print bed) were taken using a caliper to show the accuracy and precision of this manufacturing step.

The light transmission for the bottom, top, and side walls of the white reflective shell was measured in the air using a monochromatic light source and read out with an integrated sphere and a photodiode (Hamamatsu S1337-1010BQ).

**Plastic scintillator forming.** The PS fabrication consisted of three objectives: maximizing the performance of the particle detector, access to an event read out using WLS fibers in two directions (X, Y), and the preparation of the matrix-layer-in-fabrication for an additional matrix layer to be built on top.

To optimize the performance of the particle detector, the material required high transparency and a consistent fill factor throughout the voxel to maximize the active material volume. Additionally, it was important for the PS to surround the metal rods, ensuring a substantial contact area

between the scintillation material and the WLS fibers to enhance light capture efficiency. The fill density was influenced not only by the viscosity of the polymer but also by the magnitude of the mass flow rate exiting the nozzle. Increasing the flow rate resulted in a more energetic distribution of the melt within the cavity, exerting greater pressure on the material toward the boundaries. This accelerated the distribution of the melt without allowing the polymer to cool and recrystallize before reaching the outer walls of the reflective shell.

An elongated extrusion nozzle, capable of reaching the bottom of the reflective shell, facilitated an even distribution of scintillation material throughout the entire cavity. The liquid polymer was deposited through the 1.8 mm wide nozzle orifice in a vertical bottom-to-top motion, forming a single blob from which the melt spread toward the boundaries. Positioned at the top, a spring-loaded, polished stainless steel plate constrained the melt pool within the cavity (Fig. 9a). The plate incorporated a hole to enable unrestricted vertical movement of the nozzle, along with vent holes designed to release air during voxel-filling. Additionally, it featured a concave dome to provide temporary overfill, effectively compensating for any subsequent material shrinkage (Fig. 9b).

A CFD analysis using ANSYS Fluent was performed to determine the material requirements of the melting components heat block and extrusion nozzle, the geometry and material of the heat break and the feeding tube of the filament, and the process parameters heat block temperature and extrusion speed (Fig. 10a). These process elements were balanced to attain a PS outflow temperature at the nozzle orifice of around 230 °C, ensuring that its scintillation properties remained preserved[53,55], and to reach the maximum possible material throughput velocity, in order to obtain a high fill density of the PS volume by more forcefully spreading the melted polymer throughout the reflective cavity before it solidified. This was achieved by elevating the heat block temperature (above the usual working temperature) such that the PS is melted to its core at the exit point. Simultaneously, the increased heat creep from the high-temperature heat block toward the whole extrusion system needed to be controlled, such that system clogging followed by the failure of the entire injection process was prevented. The discretization of the model concluded in 321,545 nodes with a maximum polymer mesh size of 0.25 mm, a surface mesh size between one and two millimeters, three boundary layers between solid and fluid regions, and a volume mesh filled by polyhedra (Fig. 10b). Using the discretized model, the following components were determined: the filament-feeding-tube material and the heat break tube-thickness and material, both elements simulated for their heat shielding capabilities; the heat block material and temperature and

the nozzle material, both responsible for the melting of the PS. The simulation was based on a laminar flow energy model using a pressure-based transient solver. The surrounding temperature (chamber temperature of the FDM printer) was set to 50 °C, the circular cut-out in the heat block was set to a process temperature, simulating an FDM cartridge heater, and the polymer inlet velocity was varied to determine its maximum possible value. The PST-based PS was simulated using the material properties of polystyrene, with the heat break material determined to be stainless steel. The investigated material options for the feeding tube included stainless steel and polyether ether ketone, while the heat block, the coupling from heat break to heat block, and the nozzle were considered in stainless steel, copper, or brass.

The heat break is the essential element in the whole extrusion system because of its shielding function of the cold zone components from the heat creep by the heat block, thus guaranteeing a failure-free process. The performance of the manufactured heat break based on the CFD analysis was measured with a type "k" thermocouple from the manufacturer Fluke and compared to the commercially available model.

### Detector setup and analysis method for cosmic data

The $5 \times 5 \times 5$ SuperCube was instrumented with fifty readout channels, twenty-five on each view, as shown in Fig. 5a. Kuraray Y11 double-cladding 1 mm diameter WLS fibers[58] along the orthogonal X and Y directions allowed for unique identification of the cubes crossed by a charged particle. WLS fibers absorb the blue light produced by the PS cube and isotropically emit photons in the green wavelength inside their core. The different refractive indices of the outer cladding geometrically trap and guide the photons toward the SiPMs, thanks to an attenuation length of more than 4 meters. Hamamatsu Multi-Pixel Photon Counters (MPPCs) S13360-1325CS[80] were coupled with individual WLS fibers, on one of the two ends. The operating voltage was set to the one recommended by the supplier for each MPPC. The coupling was ensured by black 3D-printed optical connectors and enhanced by fixing the WLS fiber with EJ-500 optical glue[81] and pushing the MPPC toward the polished fiber end with a soft black foam, acting like a spring. The other end of the fiber was cut at 90° and polished. With a nominal photodetection efficiency of 25%, the analog charge signal of each MPPC is proportional, to the first order, to the number of photons produced in the PS cubes crossed by the corresponding WLS fiber. The charge analog signal of each MPPC was collected with micro-coaxial cables and read out by one CAEN FERS DT5202 front-end board[82], used to digitize the signal. The charge value corresponding to the highest signal peak of each independent channel was measured and converted to analog-to-digital converter (ADC) units. The number of PE was then calculated from the measured ADC units. Such conversion was obtained with a large data sample collected by exposing the prototype to a $^{90}$Sr radioactive source. The data showed a clear multi-peak structure, each one corresponding to a different number of PEs. The MPPC gain was extracted as the distance between adjacent peaks in units of ADC per PE with a multi-Gaussian distribution fit, with which the position of the first three peaks was obtained. Then, the peak positions were fitted with a linear function to extract the gain and pedestal. Since the recommended working bias voltage provided by the supplier was applied to each individual channel of the MPPC, the gain distribution was observed to be relatively uniform, averaging around 50 ADC/PE with a standard deviation of 1.21 ADC/PE. This procedure was iterated for each channel, employing its specific gain and pedestal value to convert the measured light yield into units of PE.

The performance of the SuperCube produced with FIM was compared with a reference 2D matrix of cubes produced with cast polymerization. It consisted of a single layer of $5 \times 5$ optically isolated PS 1 cm$^3$ cubes, thus a geometry analogous to the SuperCube but two dimensional. In the same way, as for the SuperCube, both orthogonal views of the reference sample were read out with Kuraray Y11 WLS fibers, coupled with the same type of MPPC following the same procedure as described above. A more detailed description of the reference prototype as well as of its particle detection performance was shown in a previous study[41]. The reference prototype read out with the same CAEN FERS DT5202 board, was placed on top of the SuperCube to trigger cosmic particles and to compare the light yield and crosstalk with the one of the SuperCube.

Both the FIM and the cast polymerization prototypes were characterized by analyzing cosmic particle events and the respective light yield and crosstalk were compared. A total of about 63 h of cosmic data was collected. Typical cosmic events were minimum ionizing particles, crossing the prototype every few seconds with an expected energy loss in PS of about 1.8 MeV cm$^{-1}$. Given their angular distribution, approximately equal to $\cos^2\theta_{azim}$, where $\theta_{azim}$ is the azimuth angle, the most common signature consisted of a vertical minimum ionizing particle leaving a track that starts from the top layer (the reference prototype) and ends at the bottom layer of the SuperCube.

The track 3D hits were reconstructed layer by layer, by determining the XY coordinate with the maximum light yield channel in both views. In each electronic channel a threshold of 300 ADC above pedestal, about 6 PE, was applied. After the track reconstruction, a data set containing only vertical tracks was selected to have a particle path length of about 1 cm in each PS cube, hence a comparable energy loss. These tracks crossed both the reference prototype and the SuperCube in the same column of cubes. The cube-to-cube light crosstalk was derived from the light yield ratio between neighboring channels within the same layer perpendicular to the track direction.

## Data availability
All the data supporting this study are available on the public repository https://doi.org/10.5281/zenodo.14228158.

## Code availability
All the codes used to analyze the data supporting this study are available on the public repository https://doi.org/10.5281/zenodo.14228158.

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

## Acknowledgements

Part of this work was supported by the SNSF grant PCEFP2_203261, Switzerland.

## Author contributions

D.S. and T.W. conceived the additive manufacturing method. T.W. implemented the method by designing and making the 3D printer components, tuning the 3D printer parameters, and producing the final 3D-printed sample. D.S. supervised the implementation process. U.K., B.L., and J.W. built the experimental setup for cosmic particle data taking. B.L. ran the particle detection experiment and performed the data analysis. C.J. and U.K. performed the measurements of reflector transmittance. U.K. and D.S. supervised the data taking and the data analysis. Also A.Ru. supervised the data analysis. A.B. and T.S. produced the scintillator filament used for the 3D-printed prototype. S.B., E.B., and S.H. provided some of the necessary equipment and discussed the results. All the authors, including A.D.R., T.D., M.F., and B.G., part of the 3D-printed DETector (3DET) R&D collaboration, discussed and commented on the manuscript.

## Competing interests

The authors declare no competing interests.
