## [Transparent Peer Review file · Communications Engineering]

Additive manufacturing of a 3D-segmented plastic scintillator detector for tracking and calorimetry of elementary particles

Corresponding Author: Professor Davide Sgalaberna

Version 0:

Reviewer comments:

Reviewer #1

(Remarks to the Author)

The article is a very well written report on a novel method for constructing a compact array of scintillation detectors with additive manufacturing methods. The procedures developed are clearly described and the obtained detector components as well as the assembly procedure are well indicated. The motivation and the limits of conventional methods of manufacturing building blocks and assembly techniques are presented and make sense to the reader. Data analysis and statistical methods utilized are understandable and reproducible. The originality is given without doubt.

I appreciate the excellent quality of the report in general and in the sense that the article reads as a technical documentation as clear as a step-by-step a How-To manual, which I find very impressive. I fully recommend the publication of the article with a few minor modifications/clarifications as requested below.

Requested modifications and corrections:

- in the last sentence of the abstract, the statement that the procedure discussed in the article would be applicable to the production of future scintillator detectors is too unspecific. I suggest to modify the statement such as "...for the production of future plastic-based scintillator detectors..." or similar in order to point out that this only refers to organic scintillation materials.
- in sec IV/B/1, second sentence of last paragraph "This material provided ...": misspelled "the"

Questions or request for clarification:

- In the introduction two material groups are indicated (PS and PVT-based scintillators), for which the proposed methods and procedures seem feasible. This raises the question, of this method would be applicable to other organic materials, such as PEN or PET, which have been shown to exhibit scintillation behavior as well. Please point out, whether your method is limited to the two beforementioned materials (and if so, for what reason) or if any extrudable material can be utilized.
- Why metal rods were placed during the molding process to define the fiber feedthroughs instead of placing the WLS fibers from the beginning? Is there a specific reason for this, such as i.e. thermal damage to the fibers? If so, mention it in the text.
- How are the WLS fiber ends processed, i.e. cutting, polishing?
- What measures were taken to prevent generation of air bubbles during the injection molding? Even if this is explained in detail in the referenced publication, it would be useful to outline this point briefly in the article.
- There are no details given to the 3d printed reflective frames. It would be useful to add a few additional technical details here, such as the fill factor of the print.
- Generally, it would be instructive to indicate the limits in terms of spatial resolution for the given SuperCube due to geometry. For a skewed track along one of the side faces for instance, the angular resolution would be sthg. like $45-39=6$ degrees (5 cubes in z and x direction respectively vs. 4 in z and 5 in x directions). If this topic is foreseen for more detailed discussion in a future publication, at least briefly mention the granularity due to the inherent geometrical limitation. For this follow-up discussion, one then would also like to discuss a possible improvement wrt this geometrical limit due to light

sharing (cross talk).

Reviewer #2

(Remarks to the Author)

The authors present a novel manufacturing technique for a 3D array of plastic scintillators for high energy physics applications. In general the work is technically sound, is an advance on the state-of-the-art, and could be impactful if the manufacturing process described can be scaled up. It is difficult to say whether this manuscript falls within the scope of the journal, in part because this journal is still relatively new. Often journals on engineering have a focus on more applied topics. However, the Aims and Scope of this journal do specifically mention “Scientific Tools and Instruments”, which would seem to include manuscripts like the one presently under consideration. In a brief search, I did not find any other papers in this journal that focus on radiation detectors. I defer to editorial judgment.

If this paper is judged to be suitable for the scope of this journal, then it potentially has the appropriate level of technical merit. However, major revisions are required:

- The introductory discussion of 3D printing in regard to plastic scintillators is unsatisfactory. In particular, the discussion in the Introduction, seventh paragraph dismisses SLA and other UV light printing as not “achieving competitive performance levels”. They cite a paper from 2014 which was the first effort of 3D printing a scintillator with light. Since then, there has been a growing body of literature on the photopolymerization and 3D printing of plastic scintillators, including the recent papers listed below. Many of these papers show good performance in light output, pulse shape discrimination, thermal neutron sensitivity, and other properties in comparison to commercially available options. Considering that, citing only one important but out-of-date paper is not sufficient evidence for the author’s claims that 3D printing via photopolymerization is not competitive. It would be more appropriate for the authors to say, e.g., that although there has been significant development in this field, the work has focused on other applications rather than HEP.
 - o Kim, et al (2023) <https://doi.org/10.1016/j.nima.2023.168537>
 - o Chandler, et al (2023) <https://doi.org/10.1016/j.addma.2023.103688>
 - o Chandler, et al (2022) <https://doi.org/10.1021/acsapm.2c00316>
 - o Frandsen, et al (2023) <https://doi.org/10.3390/jne4010019>
 - o Dolezal, et al (2023) <https://doi.org/10.1016/j.nima.2023.168602>
 - o Kim, et al (2020) <https://doi.org/10.1016/j.net.2020.05.030>
- In general the authors demonstrate too myopic a view of the field of plastic scintillators by making broad and sweeping statements that only apply to the narrow case of high energy physics. As the authors themselves point out, plastic scintillators have a wide variety of applications and many of the author’s statements that do apply to the specific case of polystyrene scintillators within high energy physics do not apply universally. The individual problematic statements are highlighted in other comments, but I feel it important to emphasize in addition that the overall tone of the paper is dismissing all non-HEP applications of plastics (i.e. the majority). Given the broad scope of this journal, the authors should be more careful with their language. Addressing the individual related minor comments below should be enough to satisfy this major comment.
- Additional characterization data is required. The authors mention that “different gains [across voxels] were found to be relatively uniform around 50 ADC/PE” but do not show any actual data or provide quantitative results. Given that the eventual goal is to make 2 million voxels, presenting the cell-to-cell variation in more detail would be quite important. This would also help make the work more reproducible. In addition, there is no measurement of the timing performance, which is an important property for this application. Lastly, how will these scintillators age in comparison to traditional polystyrene? I can understand if actual aging measurements are out of scope for this study, but there should be some discussion on what is expected based on the material properties involved. Is there any reason to expect degradation of performance (for either scintillator or reflector) based on the new technique? Several of these considerations were mentioned in the previous Berns, et al 2020 paper in JINST that has many of the same authors as this one.

In addition, the following minor revisions should be considered.

- Introduction, first paragraph: The introduction mentions a wide variety of applications for plastic scintillators. At least one citation is needed for each of them.
- Introduction, third paragraph: The sentence “Such formula has not undergone any major change in the last fifty years” is fundamentally incorrect (both grammatically and technically). Perhaps it is true that this is true for plastic scintillators in high energy physics but this is not the case more broadly. Two prominent examples are the discovery of pulse shape discrimination plastics (Zaitseva, et al <https://doi.org/10.1016/j.nima.2011.11.071>) and development of various loaded plastics (Cherepy, et al <https://doi.org/10.1016/j.nima.2015.01.008>).
- Introduction, third paragraph: The authors state that “Often, the light produced in PS is collected by wavelength-shifting (WLS) fibers.” This is not the case. Perhaps it is true for high energy physics, but not in general for plastic scintillators since most plastic scintillator applications do not use WLS fibers at all.
- Introduction, third paragraph: Is it true that the “attenuation length is maximum” for green light? Perhaps the authors mean to say that the “attenuation length is larger” for green light compared to blue/violet.
- Introduction, third paragraph: “benefiting from the over several meters low attenuation length of the fiber” should be “benefiting from the long attenuation length of the fiber, which is over several meters”
- Introduction, fourth paragraph: There are three manufacturing techniques described in this paragraph. However a large fraction of all plastic scintillators, in particular those made based on polyvinyltoluene, are thermally polymerized from a liquid

resin over many days (or even weeks). The authors should narrow their language to specifically apply to polystyrene scintillators.

- Introduction, fifth paragraph: The statement “PS detectors with tracking capabilities are made of scintillating voxels...” is too strong. Something like “A detector with optically isolated PS voxels is capable of tracking particles across multiple detector elements” would be better.
- Introduction, eighth paragraph: The authors state “the most promising additive manufacturing option is FDM.” This statement is too strong. The authors must narrow this claim to include only their application of interest.
- Introduction, eighth paragraph: The authors claim that “These scintillators are extensively utilized in particle detectors and exhibit a light yield performance comparable to the current state of the art.” Are they referring to polystyrene based scintillators in general? If so, then the second half of the sentence does not make sense because it seems like for this application polystyrene is in fact the state of the art. Are they referring to their specific scintillators from Ref [36]? If so, then they should provide citations showing that they are being “extensively utilized”.
- Fig 1. The scintillation diagram would benefit from the transition energy for the secondary fluor being lower (i.e. longer wavelength) than the one for the primary fluor. This would match with the wavelength differences shown by the arrows.
- Fig 1. I am confused about the lower right panel. First, it is not actually mentioned in the text until the Results section (after other figures have been discussed). As such it is an odd fit to put here in Fig 1. Also, is this experimental data? If so, how was it collected? Was it taken in air? If so, the indices of refraction will not be representative of the actual detector, which has no air gap.
- Results, eighth paragraph: “A transmittance of less than 10%...” should be “Although a transmittance of less than 10%...”
- Fig. 2: The middle right panel is quite blurry. Is it possible to take a better photo?
- Fig 2: The caption on the far right panel says that the surface was polished. This is at odds with the author’s claims that there is no post processing required for this technique. My understanding is that this polishing was only done for the benefit of this photo, but this could in fact be misleading to the reader who thinks that this level of transparency should be expected directly after production.
- Results, second to last paragraph: “geometrically matching the two detector readout” should be “mapping the readout channels across both boards to the corresponding geometrical elements” or something of that nature
- Fig. 3: What is the purpose of wrapping the SuperCube in PTFE? The 3D printed structure is reflective already. Does wrapping it add benefit?
- Fig 5: The right plot has large amounts of empty space with the lines of interest being quite small for much of the x axis range. A logarithmic y axis may make the figure more readable. Also, the units for the x axis should be defined in the caption. As of now they are only defined towards the very end of the paper. Lastly, are the y axis units really arbitrary? It seems like this may be a histogram showing counts.
- Discussion, sixth paragraph: The authors claim that the cube-to-cube cross talk is comparable to traditionally manufactured detectors. However, this is difficult to reconcile with the result shown in Fig 5 on the right, in which the average value of the 3D printed line is 4x higher than the average value of the standard line. It looks like there is even non negligible contribution at the 10% level in the 3D printed case. Even if the authors’ claim that the light output is high enough to make up for it is true, it does appear that the cross talk is worse.
- Discussion, ninth paragraph: The authors state that millimeter sized granularity should be possible “without problems”. However, the size of the nozzle will limit the element size. I cannot find the exact nozzle size in the paper. A smaller nozzle will require additional studies to optimize and understand the temperature of the material (as was done in this paper with CFD calculations and related experiments). In addition, smaller detectors mean less light created by minimum ionizing particles. At a certain point the light output will be too low to get a good signal. This will likely be the case well before reaching mm size with the current scintillator formulation.
- Methods, B.1, second paragraph: “performance of hte detector” should be “performance of the detector”
- Methods, B.2, second paragraph: It is odd to see the future tense when discussing that an even distribution of material “shall be achieved”. This should be in present or past tense.
- Methods, C, first paragraph: “allow to uniquely identify” should be “allow for unique identification”
- Methods, C, first paragraph: Sr90 should be 90Sr.
- Methods, C: It would be useful to see a comparison between the 90Sr calibration data for a standard cube vs a 3D printed cube. The authors discuss this data explicitly but do not show it.
- Methods, C: The sentence beginning with “After the track reconstruction...” is very difficult to parse and should be reworded for clarity. Splitting up into multiple sentences may help.
- Methods, C: As stated above, the authors claim of “slightly higher crosstalk” does not appear to be supported by evidence.

Reviewer #3

(Remarks to the Author)

This article presents the fabrication of the first additive manufactured plastic scintillator detector, capable of 3D tracking elementary particles and measuring their stopping power. Its performance is comparable to the state of the art of plastic scintillator detectors. In this work, a novel manufacturing method named Fused Injection Modeling (FIM) was developed to obtain good geometrical tolerances, high transparency PS

volumes, as well as precise hole fabrication for the placement of WLS fibers at a rapid production speed, thereby overcoming the aforementioned shortcomings of the AM fabricated PS particle detector. However, the following work still needs to be done in this work:

- 1、 Supplement some data about the properties of the plastic scintillator, such as light yield, and decay time.
- 2、 It is mentioned that the prepared plastic scintillator has sufficient transparency, but lacks data on transmittance.
- 3、 Unify the format of images in the text.

4. The results of the hydrodynamic calculations and related data are supplemented in determining the material requirements for melted parts, heat blocks and nozzles.

Version 1:

Reviewer comments:

Reviewer #1

(Remarks to the Author)

Dear Authors,

Thank You very much for the very detailed and concise explanations of the rebuttal. I went through all open issues and are fully content with the replies.

Some comments to your replies:

Thanks for your thorough comment on PEN scintillators. I agree that making statements about the feasibility of your method with this material would be presumptive.

Thanks for also clarifying the contributions to the spatial resolution.

I see all other issues or open questions settled.

I am fully satisfied with the proposed modifications and fully endorse the publication of the article in its current state.

Reviewer #2

Most of the original comments were sufficiently addressed. I appreciate the hard work by the authors to accomplish this.

Comment: In general the authors demonstrate too myopic a view of the field of plastic scintillators by making broad and sweeping statements that only apply to the narrow case of high energy physics. As the authors themselves point out, plastic scintillators have a wide variety of applications and many of the author's statements that do apply to the specific case of polystyrene scintillators within high energy physics do not apply universally. The individual problematic statements are highlighted in other comments, but I feel it important to emphasize in addition that the overall tone of the paper is dismissing all non-HEP applications of plastics (i.e. the majority). Given the broad scope of this journal, the authors should be more careful with their language. Addressing the individual related minor comments below should be enough to satisfy this major comment.

Reply: We agree that the paper lacks information related to scintillator developments in areas other than HEP, which is the focus of our research. We take note of the opinion of the referee and try to include the suggested improvements. We believe now all the comments made have been addressed (see below).

Referee Response: Most of these instances were corrected, but there remain two statements that are broad claims about plastic scintillators that only apply to high energy physics. They can be easily fixed:

- Introduction, second paragraph "PS are typically made...". This sentence should be amended to say something like "In the HEP context, PS are typically made...".

- Discussion, seventh paragraph: "It is well known that the light yield of a PS detector can be enhanced by improving the light trapping efficiency of the WLS fibers through the reduction of air gaps..." should be changed to "It is well known that the light yield of a PS detector that uses WLS fibers can be enhanced by improving the light trapping efficiency of the fibers through the reduction of air gaps."

Comment: Introduction, fourth paragraph: There are three manufacturing techniques described in this paragraph. However a large fraction of all plastic scintillators, in particular those made based on polyvinyltoluene, are thermally polymerized from a liquid resin over many days (or even weeks). The authors should narrow their language to specifically apply to polystyrene scintillators.

Reply: We do not believe this paragraph should be narrowed to specifically apply to polystyrene scintillators. In fact, the introduction aims to give a broad overview that should include the production processes of the most common plastic scintillator materials, i.e. also PVT that is mentioned at the beginning of the third paragraph. Hence, we prefer not to modify it. Moreover, the choice of a polystyrene-based plastic scintillator for 3D printing comes later in the text and it was mainly driven by the manufacturing technique, optimal for the goal achieved within this work, rather than by the type of scintillator material.

Referee Response: PVT is in fact mentioned above. However that further highlights why this paragraph should be changed. The paragraph starts "Traditionally, PSs are manufactured with..." and then continues on to list 3 techniques. This gives the reader the incorrect impression that those three techniques make up an exhaustive (or nearly so) list, which is not the case. Thermal polymerization is by far the most commonly used technique for producing commercially available plastic scintillators. This paragraph can either be fixed by adding thermal polymerization to the list or by adding a qualifier to the first sentence, e.g. "PSs for HEP experiments using fibers are traditionally manufactured by..."

Comment: Fig 2: The caption on the far right panel says that the surface was polished. This is at odds with the author's claims that there is no post processing required for this technique. My understanding is that this polishing was only done for the benefit of this photo, but this could in fact be misleading to the reader who thinks that this level of transparency should be expected directly after production.

Reply: The reflective shell was produced using FDM. FDM-fabricated structures typically exhibit surface roughness attributed to the print pattern. When the scintillation material is filled into the reflective shell, it conforms to this surface roughness. However, this imperfection in the transparency of the scintillation cube is confined solely to the interface with the reflective shell and does not adversely affect the particle detector's performance. Until this roughness is addressed, the transparency of the scintillation cube cannot be fully showcased. To be clear, this level of transparency can be expected after production throughout the 1000 mm³ volume except a few micrometers thick layer at the surface, where the PS comes in contact with the reflective shell. Moreover, as written in the caption of Fig. 2, the top layer was not covered with 3D printed reflector, thus it was not polished at all. We believe this picture gives a good idea of the level of transparency of the injected plastic scintillator, and that the caption of Fig. 2 is clear enough. A more technical explanation like the one given above.

Referee Response: A technical or confusing explanation is not needed. All that is needed is one more sentence or phrase at the end of the Fig 2 caption saying that the polishing of the bottom face was performed for illustrative purposes to show the transparency of the scintillator. A reader might otherwise get the impression that the FDM-fabricated structures are extremely flat (which is not typical, as pointed out in the reply).

Reviewer #3

(Remarks to the Author)

Revisions are appropriate. The reversed version becomes acceptable

June 12, 2024

Davide Sgalaberna
ETH Zurich, CH-8093 Zurich, Switzerland
davide.sgalaberna@cern.ch

Dear Referees,

We thank you for providing useful comments and suggestions about how to improve the manuscript. After a careful review, we believe we have addressed all of them in our replies, that you can find below.

The manuscript has been updated accordingly.

Sincerely,

Davide Sgalaberna on behalf of the authors.

Reviewer #1

The article is a very well written report on a novel method for constructing a compact array of scintillation detectors with additive manufacturing methods. The procedures developed are clearly described and the obtained detector components as well as the assembly procedure are well indicated. The motivation and the limits of conventional methods of manufacturing building blocks and assembly techniques are presented and make sense to the reader. Data analysis and statistical methods utilized are understandable and reproducible. The originality is given without doubt.

I appreciate the excellent quality of the report in general and in the sense that the article reads as a technical documentation as clear as a step-by-step a How-To manual, which i find very impressive. I fully recommend the publication of the article with a few minor modifications/clarifications as requested below.

Requested modifications and corrections:

- In the last sentence of the abstract, the statement that the procedure discussed in the article would be applicable to the production of future scintillator detectors is too unspecific. I suggest to modify the statement such as "...for the production of future plastic-based scintillator detectors..." or similar in order to point out that this only refers to organic scintillation materials.

Reply: Done.

- In sec IV/B/1, second sentence of last paragraph "This material provided ...": misspelled "the"

Reply: Done.

- In the introduction two material groups are indicated (PS and PVT-based scintillators), for which the proposed methods and procedures seem feasible. This raises the question, of this method would be applicable to other organic materials, such as PEN or PET, which have been shown to exhibit scintillation behavior as well. Please point out, wether your method is limited to the two beforementioned materials (and if so, for what reason) or if any extrudable material can be utilized.

Reply: PEN scintillator is typically manufactured through injection molding, a process that involves high temperatures similar to extrusion. Also, PEN filament is available in the market and it is produced with extrusion, like we do with our filament. Thus, we believe it should work. Therefore, in principle, we believe that the PEN scintillator should be compatible with our fabrication method. However, as we have not yet conducted experiments to validate this hypothesis, we prefer to refrain from making definitive claims in the article.

- Why metal rods were placed during the molding process to define the fiber feedthroughs instead of placing the WLS fibers from the beginning? Is there a specific reason for this, such as i.e. thermal damage to the fibers? If so, mention it in the text.

Reply: That is correct. We cannot 3D print with WLS fibers in place because they would be damaged. We initially tried an alternative solution using glass pipes which were not extracted but served as space for the WLS to be inserted after the manufacturing. To complement we added the following text in III Discussion at page 8: "An alternative option would be to utilize thin pipes capable of..."

- How are the WLS fiber ends processed, i.e. cutting, polishing?

Reply: Correct. They were cut at 90 degrees and polished. We added the following sentence in IV.C: "The other end of the fiber was cut at 90° and polished."

- What measures were taken to prevent generation of air bubbles during the injection molding? Even if this is explained in detail in the referenced publication, it would be useful to outline this point briefly in the article.

Reply: Air bubbles were prevented by the FIM method itself that keeps the injection nozzle in the melt pool at all times. Coupled with the non-turbulent outflow through the FIM nozzle, this technique prevented air from being trapped between different layers of the melted polymer. The

second paragraph of Section III - Discussions: “The FDM manufactured PS showed a sufficient. . .” has been improved with some more details about the filling procedure.

- There are no details given to the 3d printed reflective frames. It would be useful to add a few additional technical details here, such as the fill factor of the print.

Reply: The material, nozzle temperature, bed temperature, chamber temperature, layer thickness and print speed are given in the article. Since the reflective frame consists of only perimeter lines, i.e. no infill, the fill factor is 100% by definition. The paragraph “The reflective structure was fabricated using a 0.4 mm diameter nozzle with a fill factor of 100% ...” in IV.B.1 has been improved.

- Generally, it would be instructive to indicate the limits in terms of spatial resolution for the given SuperCube due to geometry. For a skewed track along one of the side faces for instance, the angular resolution would be sthg. like $45-39=6$ degrees (5 cubes in z and x direction respectively vs. 4 in z and 5 in x directions). If this topic is foreseen for more detailed discussion in a future publication, at least briefly mention the granularity due to the inherent geometrical limitation. For this follow-up discussion, one then would also like to discuss a possible improvement wrt this geometrical limit due to light sharing (cross talk).

Reply: The 1D spatial resolution is dominated by the 1cm granularity, so 3mm resolution. The impact of crosstalk is a second order effect because it is, on average, less than 4%. Moreover, being crosstalk quite low and largely stochastic, it does not help to improve the spatial resolution beyond the granularity. This has been studied by some of the authors within the T2K experiment where an analogous detector (SuperFGD, 2 tons) is currently collecting data. Results are not public yet though. On the other hand, some of the authors recently published in Commun.Phys. 6 (2023) 1, 119 the study of particle tracking resolution with both a traditional SIR particle filter and neural networks (transformer and recurrent neural network) of a large volume 3D granular scintillator detector with same granularity and comparable crosstalk. Depending on the adopted method, the 3D tracking resolution (defined as the 68% one-sided area) for a MIP ranges between 2 mm (neural networks) and 4 mm (particle filter). We believe such discussion is beyond the scope of this work. However, the following sentence has been added in the introduction: “Simulation studies conducted independently on a comparable 3D granularity plastic scintillator detector demonstrated that a tracking resolution ranging between 2 mm and 4 mm, depending on the utilized reconstruction algorithm, can be attained for minimum ionizing particles such as muons [?].”

Reviewer #2

The authors present a novel manufacturing technique for a 3D array of plastic scintillators for high energy physics applications. In general the work is technically sound, is an advance on the state-of-the-art, and could be impactful if the manufacturing process described can be scaled up. It is difficult to say whether this manuscript falls within the scope of the journal, in part because this journal is still relatively new. Often journals on engineering have a focus on more applied topics. However, the Aims and Scope of this journal do specifically mention “Scientific Tools and Instruments”, which would seem to include manuscripts like the one presently under consideration. In a brief search, I did not find any other papers in this journal that focus on radiation detectors. I defer to editorial judgment.

If this paper is judged to be suitable for the scope of this journal, then it potentially has the appropriate level of technical merit. However, major revisions are required:

- The introductory discussion of 3D printing in regard to plastic scintillators is unsatisfactory. In particular, the discussion in the Introduction, seventh paragraph dismisses SLA and other UV light printing as not “achieving competitive performance levels”. They cite a paper from 2014 which was the first effort of 3D printing a scintillator with light. Since then, there has been a growing body of literature on the photopolymerization and 3D printing of plastic scintillators, including the recent papers listed below. Many of these papers show good performance in light output, pulse shape discrimination, thermal neutron sensitivity, and other properties in comparison to commercially available options. Considering that, citing only one important but out-of-date paper is not sufficient evidence for the author’s claims that 3D printing via photopolymerization is not competitive. It would be more appropriate for the authors to say, e.g., that although there has been significant development in this field, the work has focused on other applications rather than HEP.
 - o Kim, et al (2023) <https://doi.org/10.1016/j.nima.2023.168537>
 - o Chandler, et al (2023) <https://doi.org/10.1016/j.addma.2023.103688>
 - o Chandler, et al (2022) <https://doi.org/10.1021/acsapm.2c00316>
 - o Frandsen, et al (2023) <https://doi.org/10.3390/jne4010019>
 - o Dolezal, et al (2023) <https://doi.org/10.1016/j.nima.2023.168602>
 - o Kim, et al (2020) <https://doi.org/10.1016/j.net.2020.05.030>

Reply: We apologize for missing the articles that the referee has listed. Our research focus is on high energy physics with the goal of particle tracking and calorimetry and we missed them. All the suggested references as well as the following sentence have been added in the introduction in the following sentence: “In recent years, significant progress has been made in the development of curable resins. . .”

- In general the authors demonstrate too myopic a view of the field of plastic scintillators by making broad and sweeping statements that only apply to the narrow case of high energy physics. As the authors themselves point out, plastic scintillators have a wide variety of applications and many of the author’s statements that do apply to the specific case of polystyrene scintillators within high energy physics do not apply universally. The individual problematic statements are highlighted in other comments, but I feel it important to emphasize in addition that the overall tone of the paper is dismissing all non-HEP applications of plastics (i.e. the majority). Given the broad scope of this journal, the authors should be more careful with their language. Addressing the individual related minor comments below should be enough to satisfy this major comment.

Reply: We agree that the paper lacks information related to scintillator developments in areas other than HEP, which is the focus of our research. We take note of the opinion of the referee and try to include the suggested improvements. We believe now all the comments made have been addressed (see below).

- Additional characterization data is required. The authors mention that “different gains [across voxels] were found to be relatively uniform around 50 ADC/PE” but do not show any actual data or provide quantitative results. Given that the eventual goal is to make 2 million voxels, presenting the cell-to-cell variation in more detail would be quite important. This would also help make the work more reproducible.
-

Reply: We updated the text in Sec. IV.C starting from “Since the recommended working bias voltage provided...”, adding the value of the gain uniformity and explaining more in detail the analysis procedure. Given the relatively low number of channels, we could manually set the bias voltage to the value recommended by the supplier (Hamamatsu) for each particular MPPC, which let us obtain a uniform gain. In any case, the fact that the gain/pedestal values were extracted and consistently used for each different MPPC independently allows to mitigate any systematic potential difference in the response between readout channels. This is achieved in our small setup as well as in bigger detectors, where automated calibration scripts allow one to extract and apply the gain/pedestal values for the light yield analysis of each independent channel. For instance, this is the case of the SuperFGD detector made of 2,000,000 scintillating cubes currently collecting data at the T2K neutrino experiment.

- In addition, there is no measurement of the timing performance, which is an important property for this application.

Reply: For this application, mainly focused on particle tracking and calorimetry, the time resolution is an important property, although not binding. We checked the decay time of the 3D printed scintillator and found it to be consistent with the one of UPS923A. We added a sentence in Sec. III - Discussions: “A preliminary time resolution measurement. . .”.

- Lastly, how will these scintillators age in comparison to traditional polystyrene? I can understand if actual aging measurements are out of scope for this study, but there should be some discussion on what is expected based on the material properties involved. Is there any reason to expect degradation of performance (for either scintillator or reflector) based on the new technique? Several of these considerations were mentioned in the previous Berns, et al 2020 paper in JINST that has many of the same authors as this one.

Reply: We added a short paragraph Section III - Discussions starting from: “ Aging presents another critical concern, potentially resulting in a few percent reduction in light yield annually for extruded polystyrene-based scintillators. . .” As highlighted in the text, the composition of the 3D printed scintillator is standard polystyrene, which was not the case of our first publication. Being the additive manufacturing process a pure extrusion at the standard temperature of 230 degrees, we do not expect any additional aging due to the 3D printing process that we do not already observe in plastic scintillators produced via standard methods like extrusion or injection molding. Across different cosmic data takings in a time frame of about four months, we did not observe visible variations in the scintillator light yield and crosstalk. However, more quantitative results are needed and will follow in the future with dedicated measurements.

In addition, the following minor revisions should be considered.

- Introduction, first paragraph: The introduction mentions a wide variety of applications for plastic scintillators. At least one citation is needed for each of them.

Reply: Done.

- Introduction, third paragraph: The sentence “Such formula has not undergone any major change in the last fifty years” is fundamentally incorrect (both grammatically and technically). Perhaps it is true that this is true for plastic scintillators in high energy physics but this is not the case more broadly. Two prominent examples are the discovery of pulse shape discrimination plastics (Zaitseva, et al <https://doi.org/10.1016/j.nima.2011.11.071>) and development of various loaded plastics (Cherepy, et al <https://doi.org/10.1016/j.nima.2015.01.008>).

Reply: We removed the sentence.

- Introduction, third paragraph: The authors state that “Often, the light produced in PS is collected by wavelength-shifting (WLS) fibers.” This is not the case. Perhaps it is true for high energy physics, but not in general for plastic scintillators since most plastic scintillator applications do not use WLS fibers at all.

Reply: Indeed, this is the case for large volume plastic scintillator detectors used, for example, in high-energy physics. We modified the sentence to clarify it.

-
- Introduction, third paragraph: Is it true that the “attenuation length is maximum” for green light? Perhaps the authors mean to say that the “attenuation length is larger” for green light compared to blue/violet.

Reply: It is correct. We modified the sentence.

- Introduction, third paragraph: “benefiting from the over several meters low attenuation length of the fiber” should be “benefiting from the long attenuation length of the fiber, which is over several meters”

Reply: Corrected.

- Introduction, fourth paragraph: There are three manufacturing techniques described in this paragraph. However a large fraction of all plastic scintillators, in particular those made based on polyvinyltoluene, are thermally polymerized from a liquid resin over many days (or even weeks). The authors should narrow their language to specifically apply to polystyrene scintillators.

Reply: We do not believe this paragraph should be narrowed to specifically apply to polystyrene scintillators. In fact, the introduction aims to give a broad overview that should include the production processes of the most common plastic scintillator materials, i.e. also PVT that is mentioned at the beginning of the third paragraph. Hence, we prefer not to modify it. Moreover, the choice of a polystyrene-based plastic scintillator for 3D printing comes later in the text and it was mainly driven by the manufacturing technique, optimal for the goal achieved within this work, rather than by the type of scintillator material.

- Introduction, fifth paragraph: The statement “PS detectors with tracking capabilities are made of scintillating voxels. . .” is too strong. Something like “A detector with optically isolated PS voxels is capable of tracking particles across multiple detector elements” would be better.

Reply: We modified the sentence.

- Introduction, eighth paragraph: The authors state “the most promising additive manufacturing option is FDM.” This statement is too strong. The authors must narrow this claim to include only their application of interest.

Reply: We modified the sentence.

- Introduction, eighth paragraph: The authors claim that “These scintillators are extensively utilized in particle detectors and exhibit a light yield performance comparable to the current state of the art.” Are they referring to polystyrene based scintillators in general? If so, then the second half of the sentence does not make sense because it seems like for this application polystyrene is in fact the state of the art. Are they referring to their specific scintillators from Ref [36]? If so, then they should provide citations showing that they are being “extensively utilized”.

Reply: The sentence referred to polystyrene scintillators in general. Since it is repetitive and could be misleading at this point of the text, we decided to remove it.

- Fig 1. The scintillation diagram would benefit from the transition energy for the secondary fluor being lower (i.e. longer wavelength) than the one for the primary fluor. This would match with the wavelength differences shown by the arrows.

Reply: Done.

- Fig 1. I am confused about the lower right panel. First, it is not actually mentioned in the text until the Results section (after other figures have been discussed). As such it is an odd fit to put here in Fig 1. Also, is this experimental data? If so, how was it collected? Was it taken in air? If so, the indices of refraction will not be representative of the actual detector, which has no air gap.

Reply: Although these data were collected in air, they were used as a figure of merit to optimize the 3D printing strategy of the white reflector walls. Moreover, it is still possible to compare it with measurements of standard white reflectors used in these types of detectors, like TiO₂ paint, available in literature (e.g. JINST 17 (10), P10045). We moved Fig. 1 bottom right to Annex B1 (Fig. 8) and added a paragraph at the end of that section. The caption of Fig. 8 states that the measurement was performed in air.

-
- Results, eighth paragraph: “A transmittance of less than 10%...” should be “Although a transmittance of less than 10%...”

Reply: Done.

- Fig. 2: The middle right panel is quite blurry. Is it possible to take a better photo?

Reply: Done.

- Fig 2: The caption on the far right panel says that the surface was polished. This is at odds with the author’s claims that there is no post processing required for this technique. My understanding is that this polishing was only done for the benefit of this photo, but this could in fact be misleading to the reader who thinks that this level of transparency should be expected directly after production.

Reply: The reflective shell was produced using FDM. FDM-fabricated structures typically exhibit surface roughness attributed to the print pattern. When the scintillation material is filled into the reflective shell, it conforms to this surface roughness. However, this imperfection in the transparency of the scintillation cube is confined solely to the interface with the reflective shell and does not adversely affect the particle detector’s performance. Until this roughness is addressed, the transparency of the scintillation cube cannot be fully showcased. To be clear, this level of transparency can be expected after production throughout the 1000 mm^3 volume except a few micrometers thick layer at the surface, where the PS comes in contact with the reflective shell. Moreover, as written in the caption of Fig. 2, the top layer was not covered with 3D printed reflector, thus it was not polished at all. We believe this picture gives a good idea of the level of transparency of the injected plastic scintillator, and that the caption of Fig. 2 is clear enough. A more technical explanation like the one given above might confuse the reader with too much information not relevant to the prototype performance.

- Results, second to last paragraph: “geometrically matching the two detector readout” should be “mapping the readout channels across both boards to the corresponding geometrical elements” or something of that nature

Reply: We modified the sentence. There was a missing word (“views”) that made the sentence ambiguous.

- Fig. 3: What is the purpose of wrapping the SuperCube in PTFE? The 3D printed structure is reflective already. Does wrapping it add benefit?

Reply: The SuperCube was not wrapped in PTFE. In Fig.3, the left photograph was taken before completing the 3D printing process, while the right one was taken after its completion. The caption has been updated.

- Fig 5: The right plot has large amounts of empty space with the lines of interest being quite small for much of the x axis range. A logarithmic y axis may make the figure more readable. Also, the units for the x axis should be defined in the caption. As of now they are only defined towards the very end of the paper. Lastly, are the y axis units really arbitrary? It seems like this may be a histogram showing counts.

Reply: Yes, we wrote A.U. on the y axis because both distributions have been rescaled to have a consistent normalization. We modified the plot. Now the x axis is in % to better represent the cross talk level and the y axis is in log scale now.

- Discussion, sixth paragraph: The authors claim that the cube-to-cube cross talk is comparable to traditionally manufactured detectors. However, this is difficult to reconcile with the result shown in Fig 5 on the right, in which the average value of the 3D printed line is 4x higher than the average value of the standard line. It looks like there is even non negligible contribution at the 10% level in the 3D printed case. Even if the authors’ claim that the light output is high enough to make up for it is true, it does appear that the cross talk is worse.

Reply: It is true that the reference sample showed a quite significantly lower cross-talk. However, our sentence was mainly aimed to highlighting the fact that the 3D granularity detector, which has an analogous geometry as the SuperCube 3D printed in this work and is currently collecting neutrino data at the T2K neutrino oscillation experiment in Japan, shows a cross-talk of about 3.5%. We clarified our sentence in the text citing this particular detector.

-
- Discussion, ninth paragraph: The authors state that millimeter sized granularity should be possible “without problems”. However, the size of the nozzle will limit the element size. I cannot find the exact nozzle size in the paper. A smaller nozzle will require additional studies to optimize and understand the temperature of the material (as was done in this paper with CFD calculations and related experiments).

Reply: The orifice diameter of the nozzle has been added to the text in IV.B: “The liquid polymer was deposited through the 1.8 mm wide nozzle orifice...”. It is true that to manufacture a SuperCube with finer granularity one would need to design a new nozzle. However, the goal of this paper is to show that the FIM method is reliable and can be used to deliver a SuperCube-like structure. This paper also describes the analysis method we went through to optimize the 3D printing machine and process for a particular detector geometry, allowing the authors as well as the reader to take on the same steps optimized for finer or coarser granularities.

- In addition, smaller detectors mean less light created by minimum ionizing particles. At a certain point the light output will be too low to get a good signal. This will likely be the case well before reaching mm size with the current scintillator formulation.

Reply: This is correct. For 1mm edge voxels we might expect an average of 3 p.e. / MIP, which is about 2 standard deviations away from no light detection in a single voxel. For what concerns particle tracking, in some applications it could be still desirable to sacrifice in part the light yield per single voxel with the advantage of higher spatial resolution, like for scintillating fibers. On the other hand, at the first order one expects the total light yield to be only marginally affected by the finer granularity, whilst giving the advantage of making the particle signature in the detector like a thinner track.

- Methods, B.1, second paragraph: “performance of hte detector” should be “performance of the detector”

Reply: Done.

- Methods, B.2, second paragraph: It is odd to see the future tense when discussing that an even distribution of material “shall be achieved”. This should be in present or past tense.

Reply: Done. The paragraph has been improved also to address comments from the other referees.

- Methods, C, first paragraph: “allow to uniquely identify” should be “allow for unique identification”

Reply: Done.

- Methods, C, first paragraph: Sr90 should be 90Sr.

Reply: Done.

- Methods, C: It would be useful to see a comparison between the 90Sr calibration data for a standard cube vs a 3D printed cube. The authors discuss this data explicitly but do not show it.

Reply: ⁹⁰Sr was used to calibrate SiPM not for scintillator light yield measurements. Moreover, measuring the absolute scintillation from the ⁹⁰Sr source tail is less precise than from cosmics. We prefer to not show such distribution to avoid confusion to the reader.

- Methods, C: The sentence beginning with “After the track reconstruction...” is very difficult to parse and should be reworded for clarity. Splitting up into multiple sentences may help.

Reply: Done.

- Methods, C: As stated above, the authors claim of “slightly higher crosstalk” does not appear to be supported by evidence.

Reply: See reply above.

Reviewer #3

This article presents the fabrication of the first additive manufactured plastic scintillator detector, capable of 3D tracking elementary particles and measuring their stopping power. Its performance is comparable to the state of the art of plastic scintillator detectors. In this work, a novel manufacturing method named Fused Injection Modeling (FIM) was developed to obtain good geometrical tolerances, high transparency PS volumes, as well as precise hole fabrication for the placement of WLS fibers at a rapid production speed, thereby overcoming the aforementioned shortcomings of the AM fabricated PS particle detector. However, the following work still needs to be done in this work:

- Supplement some data about the properties of the plastic scintillator, such as light yield, and decay time.

Reply: For the moment, we have not measured the absolute number of scintillation photons produced per MeV energy deposited, so we would prefer to not mention it in the article. It is difficult to extract such value from the collected data because such light output is degenerate with the scintillator attenuation length, the quality of the white reflector and the efficiency of the wavelength shifting fiber. Thus, it would necessitate additional measurements with a dedicated setup. From a rough estimate in comparison with other scintillator detectors, it is expected to be around 8,000 photons / MeV for a minimum ionizing particle, in agreement with the cited UPS 923A (Nuclear Instruments and Methods in Physics Research A 566 (2006) 286–293). However, we believe this to be beyond the scope of this article, which is to demonstrate the performance of such particular 3D printed detector geometry in particle tracking and calorimetry and to compare it with analogous state of the art detectors. For this application, mainly focused on particle tracking and calorimetry, the time resolution is an important property, although not binding. We cross checked the decay time of the 3D printed plastic scintillator and found to be consistent with the one of UPS 923A (cited in the introduction), that is 2 ns. We added a paragraph in the Section III - Discussions: “A preliminary time resolution measurement. . .”. However, we leave a more quantitative characterization to future work and publications.

- It is mentioned that the prepared plastic scintillator has sufficient transparency, but lacks data on transmittance.

Reply: In our previous publication, the technical attenuation length was measured to be approximately 19 cm using a sample produced through Fused Deposition Modeling. Upon visual inspection, it was observed that the scintillator cubes manufactured via Fused Deposition Modeling appeared less transparent compared to those produced with Fused Injection Modeling. It is worth mentioning that the plastic scintillator is exposed to the same temperature in FDM and FIM. Given that the detector granularity is only 1 cm, the attenuation length is less important than other parameters. We believe it is not within the scope of this article to provide more detailed measurements but that claiming the 19 cm attenuation length as the reference parameter value can be sufficient. This is confirmed by the measured light yield with the SuperCube.

- Unify the format of images in the text.

Reply: The final format of the images will be adjusted for the proofs.

- The results of the hydrodynamic calculations and related data are supplemented in determining the material requirements for melted parts, heat blocks and nozzles.

Reply: We do not understand what is meant by this comment. We would like to ask the referee to further clarify, if the reply given below is not satisfactory. The results of the hydrodynamic calculations (the CFD analysis) are the chosen materials and process parameters of the extrusion process that allow for:

- maintaining a consistent working temperature throughout the entire extrusion system to prevent system failure in parts that should not exceed certain temperatures
 - achieving a plastic scintillator temperature at the exit point of the extrusion system that aligns with the extrusion temperature used in previous studies
 - employing a high extrusion speed to facilitate rapid and uniform spreading of the polymer melt within the mold.
-

This is all stated in the paper. We also improved the second paragraph (“To optimize the performance. . .”) in Section IV - Methods B.2 (“Plastic scintillator forming”).

June 12, 2024

Davide Sgalaberna
ETH Zurich, CH-8093 Zurich, Switzerland
davide.sgalaberna@cern.ch

Dear Referees,

We thank you for providing useful comments and suggestions about how to improve the manuscript. After a careful review, we believe we have addressed all of them in our replies, that you can find below.

The manuscript has been updated accordingly.

Sincerely,

Davide Sgalaberna on behalf of the authors.

Reviewer #1

The article is a very well written report on a novel method for constructing a compact array of scintillation detectors with additive manufacturing methods. The procedures developed are clearly described and the obtained detector components as well as the assembly procedure are well indicated. The motivation and the limits of conventional methods of manufacturing building blocks and assembly techniques are presented and make sense to the reader. Data analysis and statistical methods utilized are understandable and reproducible. The originality is given without doubt.

I appreciate the excellent quality of the report in general and in the sense that the article reads as a technical documentation as clear as a step-by-step a How-To manual, which i find very impressive. I fully recommend the publication of the article with a few minor modifications/clarifications as requested below.

Requested modifications and corrections:

- In the last sentence of the abstract, the statement that the procedure discussed in the article would be applicable to the production of future scintillator detectors is too unspecific. I suggest to modify the statement such as "...for the production of future plastic-based scintillator detectors..." or similar in order to point out that this only refers to organic scintillation materials.

Reply: Done.

- In sec IV/B/1, second sentence of last paragraph "This material provided ...": misspelled "the"

Reply: Done.

- In the introduction two material groups are indicated (PS and PVT-based scintillators), for which the proposed methods and procedures seem feasible. This raises the question, of this method would be applicable to other organic materials, such as PEN or PET, which have been shown to exhibit scintillation behavior as well. Please point out, wether your method is limited to the two beforementioned materials (and if so, for what reason) or if any extrudable material can be utilized.

Reply: PEN scintillator is typically manufactured through injection molding, a process that involves high temperatures similar to extrusion. Also, PEN filament is available in the market and it is produced with extrusion, like we do with our filament. Thus, we believe it should work. Therefore, in principle, we believe that the PEN scintillator should be compatible with our fabrication method. However, as we have not yet conducted experiments to validate this hypothesis, we prefer to refrain from making definitive claims in the article.

- Why metal rods were placed during the molding process to define the fiber feedthroughs instead of placing the WLS fibers from the beginning? Is there a specific reason for this, such as i.e. thermal damage to the fibers? If so, mention it in the text.

Reply: That is correct. We cannot 3D print with WLS fibers in place because they would be damaged. We initially tried an alternative solution using glass pipes which were not extracted but served as space for the WLS to be inserted after the manufacturing. To complement we added the following text in III Discussion at page 8: "An alternative option would be to utilize thin pipes capable of..."

- How are the WLS fiber ends processed, i.e. cutting, polishing?

Reply: Correct. They were cut at 90 degrees and polished. We added the following sentence in IV.C: "The other end of the fiber was cut at 90° and polished."

- What measures were taken to prevent generation of air bubbles during the injection molding? Even if this is explained in detail in the referenced publication, it would be useful to outline this point briefly in the article.

Reply: Air bubbles were prevented by the FIM method itself that keeps the injection nozzle in the melt pool at all times. Coupled with the non-turbulent outflow through the FIM nozzle, this technique prevented air from being trapped between different layers of the melted polymer. The

second paragraph of Section III - Discussions: “The FDM manufactured PS showed a sufficient. . .” has been improved with some more details about the filling procedure.

- There are no details given to the 3d printed reflective frames. It would be useful to add a few additional technical details here, such as the fill factor of the print.

Reply: The material, nozzle temperature, bed temperature, chamber temperature, layer thickness and print speed are given in the article. Since the reflective frame consists of only perimeter lines, i.e. no infill, the fill factor is 100% by definition. The paragraph “The reflective structure was fabricated using a 0.4 mm diameter nozzle with a fill factor of 100% ...” in IV.B.1 has been improved.

- Generally, it would be instructive to indicate the limits in terms of spatial resolution for the given SuperCube due to geometry. For a skewed track along one of the side faces for instance, the angular resolution would be sthg. like $45-39=6$ degrees (5 cubes in z and x direction respectively vs. 4 in z and 5 in x directions). If this topic is foreseen for more detailed discussion in a future publication, at least briefly mention the granularity due to the inherent geometrical limitation. For this follow-up discussion, one then would also like to discuss a possible improvement wrt this geometrical limit due to light sharing (cross talk).

Reply: The 1D spatial resolution is dominated by the 1cm granularity, so 3mm resolution. The impact of crosstalk is a second order effect because it is, on average, less than 4%. Moreover, being crosstalk quite low and largely stochastic, it does not help to improve the spatial resolution beyond the granularity. This has been studied by some of the authors within the T2K experiment where an analogous detector (SuperFGD, 2 tons) is currently collecting data. Results are not public yet though. On the other hand, some of the authors recently published in Commun.Phys. 6 (2023) 1, 119 the study of particle tracking resolution with both a traditional SIR particle filter and neural networks (transformer and recurrent neural network) of a large volume 3D granular scintillator detector with same granularity and comparable crosstalk. Depending on the adopted method, the 3D tracking resolution (defined as the 68% one-sided area) for a MIP ranges between 2 mm (neural networks) and 4 mm (particle filter). We believe such discussion is beyond the scope of this work. However, the following sentence has been added in the introduction: “Simulation studies conducted independently on a comparable 3D granularity plastic scintillator detector demonstrated that a tracking resolution ranging between 2 mm and 4 mm, depending on the utilized reconstruction algorithm, can be attained for minimum ionizing particles such as muons [?].”

Reviewer #2

The authors present a novel manufacturing technique for a 3D array of plastic scintillators for high energy physics applications. In general the work is technically sound, is an advance on the state-of-the-art, and could be impactful if the manufacturing process described can be scaled up. It is difficult to say whether this manuscript falls within the scope of the journal, in part because this journal is still relatively new. Often journals on engineering have a focus on more applied topics. However, the Aims and Scope of this journal do specifically mention “Scientific Tools and Instruments”, which would seem to include manuscripts like the one presently under consideration. In a brief search, I did not find any other papers in this journal that focus on radiation detectors. I defer to editorial judgment.

If this paper is judged to be suitable for the scope of this journal, then it potentially has the appropriate level of technical merit. However, major revisions are required:

- The introductory discussion of 3D printing in regard to plastic scintillators is unsatisfactory. In particular, the discussion in the Introduction, seventh paragraph dismisses SLA and other UV light printing as not “achieving competitive performance levels”. They cite a paper from 2014 which was the first effort of 3D printing a scintillator with light. Since then, there has been a growing body of literature on the photopolymerization and 3D printing of plastic scintillators, including the recent papers listed below. Many of these papers show good performance in light output, pulse shape discrimination, thermal neutron sensitivity, and other properties in comparison to commercially available options. Considering that, citing only one important but out-of-date paper is not sufficient evidence for the author’s claims that 3D printing via photopolymerization is not competitive. It would be more appropriate for the authors to say, e.g., that although there has been significant development in this field, the work has focused on other applications rather than HEP.
 - o Kim, et al (2023) <https://doi.org/10.1016/j.nima.2023.168537>
 - o Chandler, et al (2023) <https://doi.org/10.1016/j.addma.2023.103688>
 - o Chandler, et al (2022) <https://doi.org/10.1021/acsapm.2c00316>
 - o Frandsen, et al (2023) <https://doi.org/10.3390/jne4010019>
 - o Dolezal, et al (2023) <https://doi.org/10.1016/j.nima.2023.168602>
 - o Kim, et al (2020) <https://doi.org/10.1016/j.net.2020.05.030>

Reply: We apologize for missing the articles that the referee has listed. Our research focus is on high energy physics with the goal of particle tracking and calorimetry and we missed them. All the suggested references as well as the following sentence have been added in the introduction in the following sentence: “In recent years, significant progress has been made in the development of curable resins. . .”

- In general the authors demonstrate too myopic a view of the field of plastic scintillators by making broad and sweeping statements that only apply to the narrow case of high energy physics. As the authors themselves point out, plastic scintillators have a wide variety of applications and many of the author’s statements that do apply to the specific case of polystyrene scintillators within high energy physics do not apply universally. The individual problematic statements are highlighted in other comments, but I feel it important to emphasize in addition that the overall tone of the paper is dismissing all non-HEP applications of plastics (i.e. the majority). Given the broad scope of this journal, the authors should be more careful with their language. Addressing the individual related minor comments below should be enough to satisfy this major comment.

Reply: We agree that the paper lacks information related to scintillator developments in areas other than HEP, which is the focus of our research. We take note of the opinion of the referee and try to include the suggested improvements. We believe now all the comments made have been addressed (see below).

- Additional characterization data is required. The authors mention that “different gains [across voxels] were found to be relatively uniform around 50 ADC/PE” but do not show any actual data or provide quantitative results. Given that the eventual goal is to make 2 million voxels, presenting the cell-to-cell variation in more detail would be quite important. This would also help make the work more reproducible.
-

Reply: We updated the text in Sec. IV.C starting from “Since the recommended working bias voltage provided...”, adding the value of the gain uniformity and explaining more in detail the analysis procedure. Given the relatively low number of channels, we could manually set the bias voltage to the value recommended by the supplier (Hamamatsu) for each particular MPPC, which let us obtain a uniform gain. In any case, the fact that the gain/pedestal values were extracted and consistently used for each different MPPC independently allows to mitigate any systematic potential difference in the response between readout channels. This is achieved in our small setup as well as in bigger detectors, where automated calibration scripts allow one to extract and apply the gain/pedestal values for the light yield analysis of each independent channel. For instance, this is the case of the SuperFGD detector made of 2,000,000 scintillating cubes currently collecting data at the T2K neutrino experiment.

- In addition, there is no measurement of the timing performance, which is an important property for this application.

Reply: For this application, mainly focused on particle tracking and calorimetry, the time resolution is an important property, although not binding. We checked the decay time of the 3D printed scintillator and found it to be consistent with the one of UPS923A. We added a sentence in Sec. III - Discussions: “A preliminary time resolution measurement. . .”.

- Lastly, how will these scintillators age in comparison to traditional polystyrene? I can understand if actual aging measurements are out of scope for this study, but there should be some discussion on what is expected based on the material properties involved. Is there any reason to expect degradation of performance (for either scintillator or reflector) based on the new technique? Several of these considerations were mentioned in the previous Berns, et al 2020 paper in JINST that has many of the same authors as this one.

Reply: We added a short paragraph Section III - Discussions starting from: “ Aging presents another critical concern, potentially resulting in a few percent reduction in light yield annually for extruded polystyrene-based scintillators. . .” As highlighted in the text, the composition of the 3D printed scintillator is standard polystyrene, which was not the case of our first publication. Being the additive manufacturing process a pure extrusion at the standard temperature of 230 degrees, we do not expect any additional aging due to the 3D printing process that we do not already observe in plastic scintillators produced via standard methods like extrusion or injection molding. Across different cosmic data takings in a time frame of about four months, we did not observe visible variations in the scintillator light yield and crosstalk. However, more quantitative results are needed and will follow in the future with dedicated measurements.

In addition, the following minor revisions should be considered.

- Introduction, first paragraph: The introduction mentions a wide variety of applications for plastic scintillators. At least one citation is needed for each of them.

Reply: Done.

- Introduction, third paragraph: The sentence “Such formula has not undergone any major change in the last fifty years” is fundamentally incorrect (both grammatically and technically). Perhaps it is true that this is true for plastic scintillators in high energy physics but this is not the case more broadly. Two prominent examples are the discovery of pulse shape discrimination plastics (Zaitseva, et al <https://doi.org/10.1016/j.nima.2011.11.071>) and development of various loaded plastics (Cherepy, et al <https://doi.org/10.1016/j.nima.2015.01.008>).

Reply: We removed the sentence.

- Introduction, third paragraph: The authors state that “Often, the light produced in PS is collected by wavelength-shifting (WLS) fibers.” This is not the case. Perhaps it is true for high energy physics, but not in general for plastic scintillators since most plastic scintillator applications do not use WLS fibers at all.

Reply: Indeed, this is the case for large volume plastic scintillator detectors used, for example, in high-energy physics. We modified the sentence to clarify it.

-
- Introduction, third paragraph: Is it true that the “attenuation length is maximum” for green light? Perhaps the authors mean to say that the “attenuation length is larger” for green light compared to blue/violet.

Reply: It is correct. We modified the sentence.

- Introduction, third paragraph: “benefiting from the over several meters low attenuation length of the fiber” should be “benefiting from the long attenuation length of the fiber, which is over several meters”

Reply: Corrected.

- Introduction, fourth paragraph: There are three manufacturing techniques described in this paragraph. However a large fraction of all plastic scintillators, in particular those made based on polyvinyltoluene, are thermally polymerized from a liquid resin over many days (or even weeks). The authors should narrow their language to specifically apply to polystyrene scintillators.

Reply: We do not believe this paragraph should be narrowed to specifically apply to polystyrene scintillators. In fact, the introduction aims to give a broad overview that should include the production processes of the most common plastic scintillator materials, i.e. also PVT that is mentioned at the beginning of the third paragraph. Hence, we prefer not to modify it. Moreover, the choice of a polystyrene-based plastic scintillator for 3D printing comes later in the text and it was mainly driven by the manufacturing technique, optimal for the goal achieved within this work, rather than by the type of scintillator material.

- Introduction, fifth paragraph: The statement “PS detectors with tracking capabilities are made of scintillating voxels. . .” is too strong. Something like “A detector with optically isolated PS voxels is capable of tracking particles across multiple detector elements” would be better.

Reply: We modified the sentence.

- Introduction, eighth paragraph: The authors state “the most promising additive manufacturing option is FDM.” This statement is too strong. The authors must narrow this claim to include only their application of interest.

Reply: We modified the sentence.

- Introduction, eighth paragraph: The authors claim that “These scintillators are extensively utilized in particle detectors and exhibit a light yield performance comparable to the current state of the art.” Are they referring to polystyrene based scintillators in general? If so, then the second half of the sentence does not make sense because it seems like for this application polystyrene is in fact the state of the art. Are they referring to their specific scintillators from Ref [36]? If so, then they should provide citations showing that they are being “extensively utilized”.

Reply: The sentence referred to polystyrene scintillators in general. Since it is repetitive and could be misleading at this point of the text, we decided to remove it.

- Fig 1. The scintillation diagram would benefit from the transition energy for the secondary fluor being lower (i.e. longer wavelength) than the one for the primary fluor. This would match with the wavelength differences shown by the arrows.

Reply: Done.

- Fig 1. I am confused about the lower right panel. First, it is not actually mentioned in the text until the Results section (after other figures have been discussed). As such it is an odd fit to put here in Fig 1. Also, is this experimental data? If so, how was it collected? Was it taken in air? If so, the indices of refraction will not be representative of the actual detector, which has no air gap.

Reply: Although these data were collected in air, they were used as a figure of merit to optimize the 3D printing strategy of the white reflector walls. Moreover, it is still possible to compare it with measurements of standard white reflectors used in these types of detectors, like TiO₂ paint, available in literature (e.g. JINST 17 (10), P10045). We moved Fig. 1 bottom right to Annex B1 (Fig. 8) and added a paragraph at the end of that section. The caption of Fig. 8 states that the measurement was performed in air.

-
- Results, eighth paragraph: “A transmittance of less than 10%...” should be “Although a transmittance of less than 10%...”

Reply: Done.

- Fig. 2: The middle right panel is quite blurry. Is it possible to take a better photo?

Reply: Done.

- Fig 2: The caption on the far right panel says that the surface was polished. This is at odds with the author’s claims that there is no post processing required for this technique. My understanding is that this polishing was only done for the benefit of this photo, but this could in fact be misleading to the reader who thinks that this level of transparency should be expected directly after production.

Reply: The reflective shell was produced using FDM. FDM-fabricated structures typically exhibit surface roughness attributed to the print pattern. When the scintillation material is filled into the reflective shell, it conforms to this surface roughness. However, this imperfection in the transparency of the scintillation cube is confined solely to the interface with the reflective shell and does not adversely affect the particle detector’s performance. Until this roughness is addressed, the transparency of the scintillation cube cannot be fully showcased. To be clear, this level of transparency can be expected after production throughout the 1000 mm^3 volume except a few micrometers thick layer at the surface, where the PS comes in contact with the reflective shell. Moreover, as written in the caption of Fig. 2, the top layer was not covered with 3D printed reflector, thus it was not polished at all. We believe this picture gives a good idea of the level of transparency of the injected plastic scintillator, and that the caption of Fig. 2 is clear enough. A more technical explanation like the one given above might confuse the reader with too much information not relevant to the prototype performance.

- Results, second to last paragraph: “geometrically matching the two detector readout” should be “mapping the readout channels across both boards to the corresponding geometrical elements” or something of that nature

Reply: We modified the sentence. There was a missing word (“views”) that made the sentence ambiguous.

- Fig. 3: What is the purpose of wrapping the SuperCube in PTFE? The 3D printed structure is reflective already. Does wrapping it add benefit?

Reply: The SuperCube was not wrapped in PTFE. In Fig.3, the left photograph was taken before completing the 3D printing process, while the right one was taken after its completion. The caption has been updated.

- Fig 5: The right plot has large amounts of empty space with the lines of interest being quite small for much of the x axis range. A logarithmic y axis may make the figure more readable. Also, the units for the x axis should be defined in the caption. As of now they are only defined towards the very end of the paper. Lastly, are the y axis units really arbitrary? It seems like this may be a histogram showing counts.

Reply: Yes, we wrote A.U. on the y axis because both distributions have been rescaled to have a consistent normalization. We modified the plot. Now the x axis is in % to better represent the cross talk level and the y axis is in log scale now.

- Discussion, sixth paragraph: The authors claim that the cube-to-cube cross talk is comparable to traditionally manufactured detectors. However, this is difficult to reconcile with the result shown in Fig 5 on the right, in which the average value of the 3D printed line is 4x higher than the average value of the standard line. It looks like there is even non negligible contribution at the 10% level in the 3D printed case. Even if the authors’ claim that the light output is high enough to make up for it is true, it does appear that the cross talk is worse.

Reply: It is true that the reference sample showed a quite significantly lower cross-talk. However, our sentence was mainly aimed to highlighting the fact that the 3D granularity detector, which has an analogous geometry as the SuperCube 3D printed in this work and is currently collecting neutrino data at the T2K neutrino oscillation experiment in Japan, shows a cross-talk of about 3.5%. We clarified our sentence in the text citing this particular detector.

-
- Discussion, ninth paragraph: The authors state that millimeter sized granularity should be possible “without problems”. However, the size of the nozzle will limit the element size. I cannot find the exact nozzle size in the paper. A smaller nozzle will require additional studies to optimize and understand the temperature of the material (as was done in this paper with CFD calculations and related experiments).

Reply: The orifice diameter of the nozzle has been added to the text in IV.B: “The liquid polymer was deposited through the 1.8 mm wide nozzle orifice...”. It is true that to manufacture a SuperCube with finer granularity one would need to design a new nozzle. However, the goal of this paper is to show that the FIM method is reliable and can be used to deliver a SuperCube-like structure. This paper also describes the analysis method we went through to optimize the 3D printing machine and process for a particular detector geometry, allowing the authors as well as the reader to take on the same steps optimized for finer or coarser granularities.

- In addition, smaller detectors mean less light created by minimum ionizing particles. At a certain point the light output will be too low to get a good signal. This will likely be the case well before reaching mm size with the current scintillator formulation.

Reply: This is correct. For 1mm edge voxels we might expect an average of 3 p.e. / MIP, which is about 2 standard deviations away from no light detection in a single voxel. For what concerns particle tracking, in some applications it could be still desirable to sacrifice in part the light yield per single voxel with the advantage of higher spatial resolution, like for scintillating fibers. On the other hand, at the first order one expects the total light yield to be only marginally affected by the finer granularity, whilst giving the advantage of making the particle signature in the detector like a thinner track.

- Methods, B.1, second paragraph: “performance of hte detector” should be “performance of the detector”

Reply: Done.

- Methods, B.2, second paragraph: It is odd to see the future tense when discussing that an even distribution of material “shall be achieved”. This should be in present or past tense.

Reply: Done. The paragraph has been improved also to address comments from the other referees.

- Methods, C, first paragraph: “allow to uniquely identify” should be “allow for unique identification”

Reply: Done.

- Methods, C, first paragraph: Sr90 should be 90Sr.

Reply: Done.

- Methods, C: It would be useful to see a comparison between the 90Sr calibration data for a standard cube vs a 3D printed cube. The authors discuss this data explicitly but do not show it.

Reply: ⁹⁰Sr was used to calibrate SiPM not for scintillator light yield measurements. Moreover, measuring the absolute scintillation from the ⁹⁰Sr source tail is less precise than from cosmics. We prefer to not show such distribution to avoid confusion to the reader.

- Methods, C: The sentence beginning with “After the track reconstruction...” is very difficult to parse and should be reworded for clarity. Splitting up into multiple sentences may help.

Reply: Done.

- Methods, C: As stated above, the authors claim of “slightly higher crosstalk” does not appear to be supported by evidence.

Reply: See reply above.

Reviewer #3

This article presents the fabrication of the first additive manufactured plastic scintillator detector, capable of 3D tracking elementary particles and measuring their stopping power. Its performance is comparable to the state of the art of plastic scintillator detectors. In this work, a novel manufacturing method named Fused Injection Modeling (FIM) was developed to obtain good geometrical tolerances, high transparency PS volumes, as well as precise hole fabrication for the placement of WLS fibers at a rapid production speed, thereby overcoming the aforementioned shortcomings of the AM fabricated PS particle detector. However, the following work still needs to be done in this work:

- Supplement some data about the properties of the plastic scintillator, such as light yield, and decay time.

Reply: For the moment, we have not measured the absolute number of scintillation photons produced per MeV energy deposited, so we would prefer to not mention it in the article. It is difficult to extract such value from the collected data because such light output is degenerate with the scintillator attenuation length, the quality of the white reflector and the efficiency of the wavelength shifting fiber. Thus, it would necessitate additional measurements with a dedicated setup. From a rough estimate in comparison with other scintillator detectors, it is expected to be around 8,000 photons / MeV for a minimum ionizing particle, in agreement with the cited UPS 923A (Nuclear Instruments and Methods in Physics Research A 566 (2006) 286–293). However, we believe this to be beyond the scope of this article, which is to demonstrate the performance of such particular 3D printed detector geometry in particle tracking and calorimetry and to compare it with analogous state of the art detectors. For this application, mainly focused on particle tracking and calorimetry, the time resolution is an important property, although not binding. We cross checked the decay time of the 3D printed plastic scintillator and found to be consistent with the one of UPS 923A (cited in the introduction), that is 2 ns. We added a paragraph in the Section III - Discussions: “A preliminary time resolution measurement. . .”. However, we leave a more quantitative characterization to future work and publications.

- It is mentioned that the prepared plastic scintillator has sufficient transparency, but lacks data on transmittance.

Reply: In our previous publication, the technical attenuation length was measured to be approximately 19 cm using a sample produced through Fused Deposition Modeling. Upon visual inspection, it was observed that the scintillator cubes manufactured via Fused Deposition Modeling appeared less transparent compared to those produced with Fused Injection Modeling. It is worth mentioning that the plastic scintillator is exposed to the same temperature in FDM and FIM. Given that the detector granularity is only 1 cm, the attenuation length is less important than other parameters. We believe it is not within the scope of this article to provide more detailed measurements but that claiming the 19 cm attenuation length as the reference parameter value can be sufficient. This is confirmed by the measured light yield with the SuperCube.

- Unify the format of images in the text.

Reply: The final format of the images will be adjusted for the proofs.

- The results of the hydrodynamic calculations and related data are supplemented in determining the material requirements for melted parts, heat blocks and nozzles.

Reply: We do not understand what is meant by this comment. We would like to ask the referee to further clarify, if the reply given below is not satisfactory. The results of the hydrodynamic calculations (the CFD analysis) are the chosen materials and process parameters of the extrusion process that allow for:

- maintaining a consistent working temperature throughout the entire extrusion system to prevent system failure in parts that should not exceed certain temperatures
 - achieving a plastic scintillator temperature at the exit point of the extrusion system that aligns with the extrusion temperature used in previous studies
 - employing a high extrusion speed to facilitate rapid and uniform spreading of the polymer melt within the mold.
-

This is all stated in the paper. We also improved the second paragraph (“To optimize the performance. . .”) in Section IV - Methods B.2 (“Plastic scintillator forming”).

December 14, 2024

Davide Sgalaberna
ETH Zurich, CH-8093 Zurich, Switzerland
davide.sgalaberna@cern.ch

Dear Reviewers,

We thank you for providing useful comments and suggestions about how to improve the manuscript.
Also the comments from Reviewer #2 are now addressed. The manuscript has been updated accordingly.

Sincerely,

Davide Sgalaberna on behalf of the authors.

Reviewer #1

Dear Authors, Thank You very much for the very detailed and concise explanations of the rebuttal. I went through all open issues and are fully content with the replies. Some comments to your replies: Thanks for your thorough comment on PEN scintillators. I agree that making statements about the feasibility of your method with this material would be presumptive. Thanks for also clarifying the contributions to the spatial resolution. I see all other issues or open questions settled. I am fully satisfied with the proposed modifications and fully endorse the publication of the article in its current state.

Reviewer #3

Revisions are appropriate. The reversed version becomes acceptable

Reviewer #2

Changes in the manuscript were made accordingly.

Most of the original comments were sufficiently addressed. I appreciate the hard work by the authors to accomplish this.

Comment: In general the authors demonstrate too myopic a view of the field of plastic scintillators by making broad and sweeping statements that only apply to the narrow case of high energy physics. As the authors themselves point out, plastic scintillators have a wide variety of applications and many of the author's statements that do apply to the specific case of polystyrene scintillators within high energy physics do not apply universally. The individual problematic statements are highlighted in other comments, but I feel it important to emphasize in addition that the overall tone of the paper is dismissing all non-HEP applications of plastics (i.e. the majority). Given the broad scope of this journal, the authors should be more careful with their language. Addressing the individual related minor comments below should be enough to satisfy this major comment.

Reply: We agree that the paper lacks information related to scintillator developments in areas other than HEP, which is the focus of our research. We take note of the opinion of the referee and try to include the suggested improvements. We believe now all the comments made have been addressed (see below).

Referee Response: Most of these instances were corrected, but there remain two statements that are broad claims about plastic scintillators that only apply to high energy physics. They can be easily fixed: - Introduction, second paragraph "PS are typically made...". This sentence should be amended to say something like "In the HEP context, PS are typically made...". - Discussion, seventh paragraph: "It is well known that the light yield of a PS detector can be enhanced by improving the light trapping efficiency of the WLS fibers through the reduction of air gaps..." should be changed to "It is well known that the light yield of a PS detector that uses WLS fibers can be enhanced by improving the light trapping efficiency of the fibers through the reduction of air gaps."

Reply by the authors: this has been addressed in the latest version of the manuscript.

Comment: Introduction, fourth paragraph: There are three manufacturing techniques described in this paragraph. However a large fraction of all plastic scintillators, in particular those made based on polyvinyltoluene, are thermally polymerized from a liquid resin over many days (or even weeks). The authors should narrow their language to specifically apply to polystyrene scintillators.

Reply: We do not believe this paragraph should be narrowed to specifically apply to polystyrene scintillators. In fact, the introduction aims to give a broad overview that should include the production processes of the most common plastic scintillator materials, i.e. also PVT that is mentioned at the beginning of the

third paragraph. Hence, we prefer not to modify it. Moreover, the choice of a polystyrene-based plastic scintillator for 3D printing comes later in the text and it was mainly driven by the manufacturing technique, optimal for the goal achieved within this work, rather than by the type of scintillator material.

Referee Response: PVT is in fact mentioned above. However that further highlights why this paragraph should be changed. The paragraph starts “Traditionally, PSs are manufactured with. . .” and then continues on to list 3 techniques. This gives the reader the incorrect impression that those three techniques make up an exhaustive (or nearly so) list, which is not the case. Thermal polymerization is by far the most commonly used technique for producing commercially available plastic scintillators. This paragraph can either be fixed by adding thermal polymerization to the list or by adding a qualifier to the first sentence, e.g. “PSs for HEP experiments using fibers are traditionally manufactured by. . .”

Reply by the authors: this has been addressed in the latest version of the manuscript.

Comment: Fig 2: The caption on the far right panel says that the surface was polished. This is at odds with the author’s claims that there is no post processing required for this technique. My understanding is that this polishing was only done for the benefit of this photo, but this could in fact be misleading to the reader who thinks that this level of transparency should be expected directly after production.

Reply: The reflective shell was produced using FDM. FDM-fabricated structures typically exhibit surface roughness attributed to the print pattern. When the scintillation material is filled into the reflective shell, it conforms to this surface roughness. However, this imperfection in the transparency of the scintillation cube is confined solely to the interface with the reflective shell and does not adversely affect the particle detector’s performance. Until this roughness is addressed, the transparency of the scintillation cube cannot be fully showcased. To be clear, this level of transparency can be expected after production throughout the 1000 mm³ volume except a few micrometers thick layer at the surface, where the PS comes in contact with the reflective shell. Moreover, as written in the caption of Fig. 2, the top layer was not covered with 3D printed reflector, thus it was not polished at all. We believe this picture gives a good idea of the level of transparency of the injected plastic scintillator, and that the caption of Fig. 2 is clear enough. A more technical explanation like the one given above.

Referee Response: A technical or confusing explanation is not needed. All that is needed is one more sentence or phrase at the end of the Fig 2 caption saying that the polishing of the bottom face was performed for illustrative purposes to show the transparency of the scintillator. A reader might otherwise get the impression that the FDM-fabricated structures are extremely flat (which is not typical, as pointed out in the reply).

Reply by the authors: this has been addressed in the latest version of the manuscript.